# Self-regenerating giant hyaluronan polymer brushes

Wenbin Wei [1,8], Jessica L. Faubel [1,8], Hemaa Selvakumar[1,2], Daniel T. Kovari [1,3], Joanna Tsao[4], Felipe Rivas[5], Amar T. Mohabir[4], Michelle Krecker[6], Elaheh Rahbar [5], Adam R. Hall[5], Michael A. Filler [4], Jennifer L. Washburn [7], Paul H. Weigel[7] & Jennifer E. Curtis [1,2]*

Tailoring interfaces with polymer brushes is a commonly used strategy to create functional materials for numerous applications. Existing methods are limited in brush thickness, the ability to generate high-density brushes of biopolymers, and the potential for regeneration. Here we introduce a scheme to synthesize ultra-thick regenerating hyaluronan polymer brushes using hyaluronan synthase. The platform provides a dynamic interface with tunable brush heights that extend up to 20 microns – two orders of magnitude thicker than standard brushes. The brushes are easily sculpted into micropatterned landscapes by photo-deactivation of the enzyme. Further, they provide a continuous source of megadalton hyaluronan or they can be covalently-stabilized to the surface. Stabilized brushes exhibit superb resistance to biofilms, yet are locally digested by fibroblasts. This brush technology provides opportunities in a range of arenas including regenerating tailorable biointerfaces for implants, wound healing or lubrication as well as fundamental studies of the glycocalyx and polymer physics.

[1] School of Physics, Georgia Institute of Technology, Atlanta, GA, USA. [2] Petit H. Parker Institute for Bioengineering and Bioscience, Georgia Institute of Technology, Atlanta, GA, USA. [3] Department of Physics, Emory University, Atlanta, GA, USA. [4] School of Chemical and Biomolecular Engineering, Georgia Institute of Technology, Atlanta, GA, USA. [5] Virginia Tech-Wake Forest University School of Biomedical Engineering and Sciences, Wake Forest School of Medicine, Winston-Salem, NC, USA. [6] School of Materials Science and Engineering, Georgia Institute of Technology, Atlanta, GA, USA. [7] Department of Biochemistry and Molecular Biology, University of Oklahoma Health Sciences Center, Oklahoma City, OK, USA. [8] These authors contributed equally: Wenbin Wei, Jessica L. Faubel *email: jennifer.curtis@physics.gatech.edu

The design of functional biomaterials requires exquisite control over interfaces. Integrating physical and chemical features that can be dynamically tuned such as self-healing, stimulus responsiveness, stiffness, crosslinking, or hydrophobicity is a common goal in many modern biomaterials applications[1–9]. One popular strategy to engineer such sophisticated biointerfaces is the polymer brush[10]. Polymer brushes have been used to optimize a range of applications including sensors[11], anti-biofouling coatings[12–14], protein separations[15], drug delivery[16], tissue engineering[17,18] and implants[19,20]. The brushes used in these applications are typically high- density, and therefore require assembly using a grafting-from strategy rather than a grafting-to approach. This constraint is unfortunate since natural biopolymers are often desirable for biomaterials applications, yet few methodologies exist for the synthetic catalysis of biopolymers from surfaces. Further, many synthetic polymer brushes carry the disadvantage that they cannot be dynamically grown in vivo or regenerated after wear. This work overcomes these issues by introducing a versatile platform to grow ultra-thick, dense hyaluronan polymer brushes, whose biocompatibility and possibility for regeneration provides a route for controlling dynamic biointerfaces.

The biopolymer hyaluronan (HA, also called hyaluronic acid) is widely used in an array of biomedical and clinical applications[21,22]. Ubiquitous throughout human tissues and fluids, HA is a nonimmunogenic polysaccharide comprised of alternating N-acetyl-D-glucosamine (GlcNAc) and D-glucuronic acid (GlcUA) (Fig. 1a)[23,24]. It is essential for a multitude of cellular and tissue functions. The role of HA in the body depends on its size, which can range from a few oligosaccharides (1 disaccharide is 400 Da and corresponds to ~1 nm) to up to at least 25,000 disaccharide units (10 MDa, 25 μm). HA is a common component in biomaterials, cosmetics and therapeutics due to its biocompatibility, its ease of chemical modification, its hydrophilic properties, and its large size, which when combined with crosslinking endows valuable viscoelastic properties and the capacity to form extremely hydrated matrices[25,26].

Man-made HA polymer brushes are rare and limited by a grafting-to approach[27,28]. Yet nature generates HA polymer brush-like structures in the cell glycocalyx[29]. In organisms ranging from algae to bacteria to vertebrates, HA is assembled and retained at the cell surface by the enzyme hyaluronan synthase (HA synthase)[30]. This transmembrane protein manages the dual tasks of polymerizing as well as translocating HA across the cell membrane at ~1 nm per s into the extracellular space (Fig. 1b)[31,32]. HA synthase, equipped with two glycotransferase sites, utilizes the uridine diphosphate sugar substrates (UDP-GlcUA and UDP-GlcNAc) in the presence of $Mg^{2+}$ to polymerize the HA[33,34]. The final HA product often remains bound to the HA synthase even after synthesis stops.

In this work, we leverage the capacity of engineered bacteria to express high densities of HA synthase in their membranes, as well as the ability of HA synthase to generate extraordinarily long polymers. Immobilizing HA synthase-rich bacterial membrane fragments on interfaces provides a methodology to generate tunable, regenerative, giant HA brushes driven by enzymatic HA synthesis. With HA brush heights in the micron range fabricated in less than an hour, the brushes have the distinct advantage that they can be directly visualized with fluorescence microscopy. This platform is an original technology for growing brushes, using neither the traditional grafting-to nor grafting-from methodology. The outcome is one of the thickest (~22 μm, Fig. 1), if not the thickest polymer brush ever reported for synthetic or natural polymers. Here we introduce how to make the active enzymatic interfaces. We then characterize the resultant brush structure, demonstrate its regenerative properties, engineer the brush interface to expand performance range (stabilization, micro-patterning), and explore a range of potential applications in fundamental and applied materials science.

## Results

**Immobilization of hyaluronan synthase membrane fragments.** Hyaluronan synthase activity is robustly preserved in fragments of bacterial membrane. The activity of the HA synthase within the membranes can be dynamically switched on and off by controlling the availability of $Mg^{2+}$ or the UDP-sugars used by the enzyme to make HA[35,36]. Previous studies used HA synthase-rich membrane fragments to investigate the function and kinetics of HA synthase, including the Group C *Streptococcus equisimilus* HA synthase used in this work[31,35]. The work also extensively characterized the molecular weight distribution produced by the HAS fragments using SEC-MALLS (size exclusion chromatography—multi angle laser light scattering). Those studies show the HAS fragments produce a wide distribution of HA sizes, whose average molecular weight increases with time and plateaus at 4–8 h with an average size of ~6 MDa ($M_w$).

Membrane fragment immobilization onto surfaces is achieved by first coating glass surfaces with polyethyleneimine (PEI) and then activating with glutaraldehyde (GA). The primed surface is used to crosslink with the numerous proteins in the membrane fragments (Fig. 1c). Surprisingly, despite the underlying glass interface, our data demonstrate that the UDP-sugar monomers must have access to the enzymes, which still manage to rapidly polymerize and extrude HA into the surrounding area. Indeed, the resultant HA at the interface is so extensive that it visibly excludes 200 nm particles from the underlying surface and establishes a giant gap of >20 μm in low ionic strength conditions (Fig. 1g, $t_{growth} = 16$ h).

We used solid-state nanopore sensor technology[37] to characterize the output of the HA enzyme by the membrane fragments over time. These data are complementary to our already published data from SEC-MALLS, but are more accurate because of the sensor's ability to measure high molecular weight polymers and the full molecular weight distribution. A snapshot of the HA molecular weight distributions at 1, 2, and 8 h of synthesis is shown in Fig. 1f (see Supplementary Table 1). The resulting HA is polydisperse, spanning two orders of magnitude. At 8 h, the distribution has an average contour length of ~12.8 μm, corresponding to 5.13 MDa ($M_w$). Notably, much longer chains were detected (up to 20 MDa). The average $M_w$ and $M_n$ versus growth time are summarized in Table 1 in the Supplementary Information. We also characterized the membrane fragment size with dynamic light scattering and scanning electron microscopy (SEM). Light scattering measurements yield a diameter of ~145 nm (before HA synthesis), consistent with SEM imaging (Fig. 1d, e; see Supplementary Fig. 1).

To estimate the grafting density of the HA polymer brushes, we measured the density of HA synthase in the membrane fragments. Typical bacterial membranes have a density of 30,000 proteins per $\mu m^2$ [38]. Protein gel electrophoresis analysis of the HA synthase-enriched membrane fragments indicates that 6.8% of membrane proteins by weight are HA synthase (see Supplementary Fig. 2). This yields an estimate for HA synthase density of 2040 molecules per $\mu m^2$ (0.002 chains per $nm^2$) corresponding to a ~22 nm spacing in the membrane fragments. Further, the membrane fragments cover an estimated 30% of the surface with an average spacing of a few hundred nanometers as determined by SEM.

Measurements of the grafting density extracted from the height of the dried HA brush (4 h) compare well with the above estimate. The dry brush height was measured to be $12.5 \pm 0.7$ nm

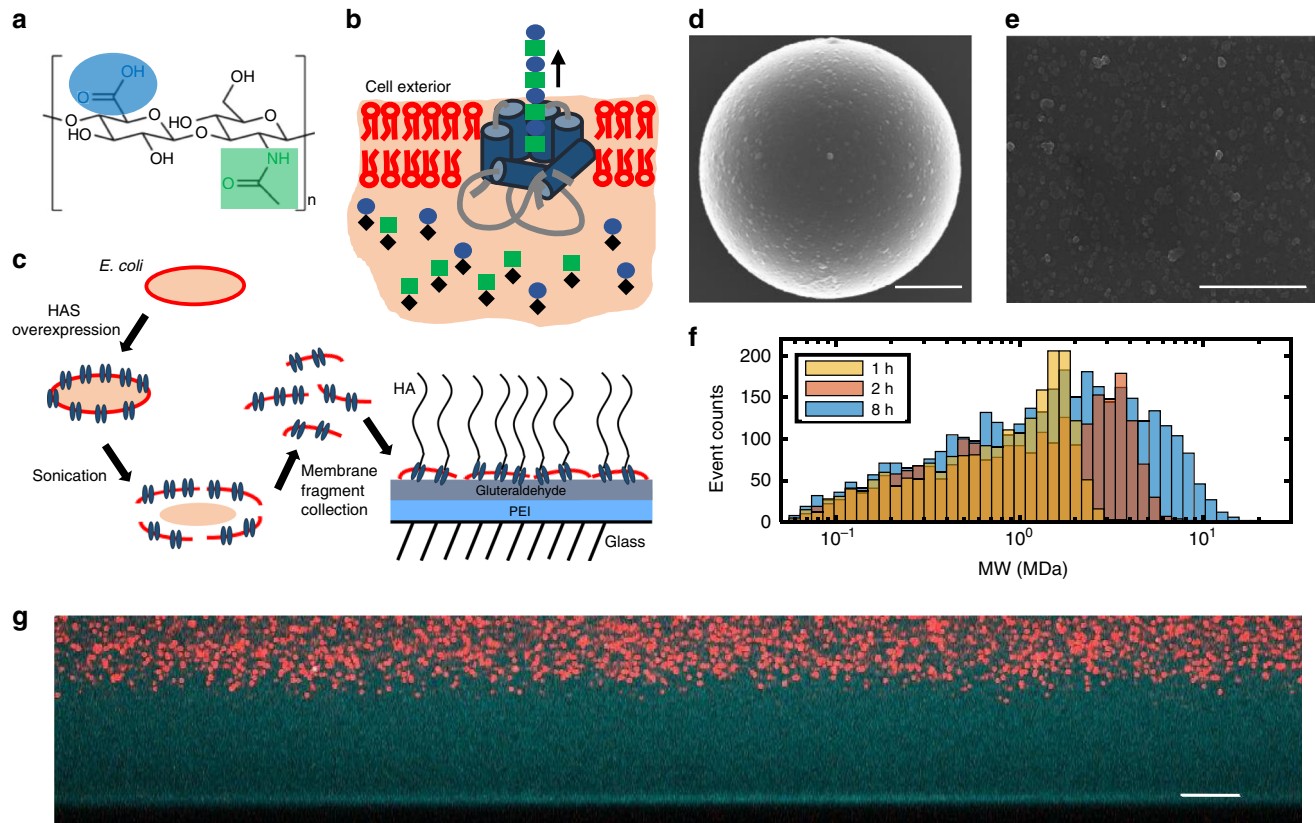

**Fig. 1** Construction of the hyaluronan synthase brush. **a** The disaccharide unit of HA is comprised of D-glucuronic acid (GlcUA) and N-acetyl-D-glucosamine (GlcNAc). **b** HA synthase (blue) embedded in a bacterial membrane (red) polymerizes and extrudes the growing HA polymer through the pore. The sugar substrates UDP-GlcUA (circle) and UDP-GlcNAc (square) are alternatively bound at the intracellular glycotransferase sites where processive assembly takes place. HA is extruded at an average rate of ~1 nm/s corresponding to 1 disaccharide/s. **c** Preparation and immobilization of membrane fragments carrying HA synthase to glass substrates. **d**, **e** SEM images of the membrane fragments on spherical and planar surfaces. Scale bars 2 μm. **f** HA molecular weight distributions assayed by solid-state nanopore ($N = 2091$, 1 h; $N = 2500$, 2 h; $N = 3699$, 8 h). See Supplementary Info, Table 1. **g** Side view confocal image of HA brush grown for 16 h and imaged at low ionic strength (1.5 mM). The brush region is imaged by the contrast generated by its accessibility to fluorescent dextran (cyan, 10 kDa), but exclusion of nanoparticles (red, 200 nm). The brush height is ~22 μm. Scale bar 10 μm.

($n = 3$, ± is st. dev.) using atomic force microscopy (AFM, see Supplementary Fig. 3). The grafting density can be found using the relationship $H = N \tau^{-1} \sigma b^3$, where $H$ = dry brush thickness, $N$ = number of monomers, $b$ = monomer length (1 nm for HA), $\sigma$ is the grafting density, and $\tau$ is the second virial coefficient indicating the solvent quality ($\tau = 1$ for poor solvent)[39].

To estimate the monomer number, $N$, for a polydisperse system at $t_{growth} = 4$ h, we can use the number-averaged molecular weight $M_{n}$, 2.42 MDa as measured by the solid-state nanopore sensor technology, which yields $N = 6050$; or we can use the weight-averaged molecular weight $M_{w}$, 6.76 MDa, which yields $N = 16,900$. Hence, examining a range of $6050 < N < 16,900$ and $H = 12.5$ nm, we arrive at a grafting density range of 0.00074–0.0021 chains per nm², or equivalently, 740–2100 chains per μm². Notably, these values are two orders of magnitude smaller than those for a typical synthetic brush generated via controlled radical polymerization (typically ~0.1 to 1 chain per nm²). However, the density is still effectively high for this system, considering the micrometer lengths of the grafted polymers. Since we expect that about 50% of the fragments are upside down and only 30% of the surface is covered by HAS fragments, an approximate grafting density of 740 chains per μm² as compared to a maximum 2100 HAS density in the fragments (predicted from protein analysis) seems reasonable, as it is 35% of the estimated enzyme density.

**The HA synthase brush**. The ultra-thick HA synthase-generated brushes afford unusual access to spatial characterization via high-resolution confocal microscopy. Figure 2a, b shows cross-sections of the fluorescently labeled HA brushes grown on a planar glass surface and of spherical brushes grown on 8 μm glass microspheres. For both geometries, the growth took place for 4 h at 30 °C ($t_{growth} = 4$ h). The concentration profiles of both planar and spherical brushes are well fit by an exponential (see the distribution of fitting parameters shown in Supplementary Note 1 and Supplementary Fig. 4). A monodisperse planar brush is predicted to have a parabolic concentration profile[40] while polydisperse brushes, both planar and spherical, have a profile that depends on their length distribution. Hence the nonparabolic profile is consistent with the polydispersity associated with HA synthase (Fig. 1f, Supplementary Table 1)[41,42].

That HA polymers are indeed produced in this unusual configuration was verified in several ways. First, we demonstrated that the enzyme Streptomyces hyaluronidase, which is highly specific for HA[43,44], quickly and thoroughly degrades the polymer brushes. This result was further confirmed by the specific binding of HA-binding proteins tagged with GFP fluorophores used to make the fluorescent profiles of the brush shown in Fig. 2a, b[45,46]. Last, we performed X-ray photoelectron spectroscopy (XPS) measurements of the dried brushes. Although the resultant spectra were a convolution from the underlying PEI/

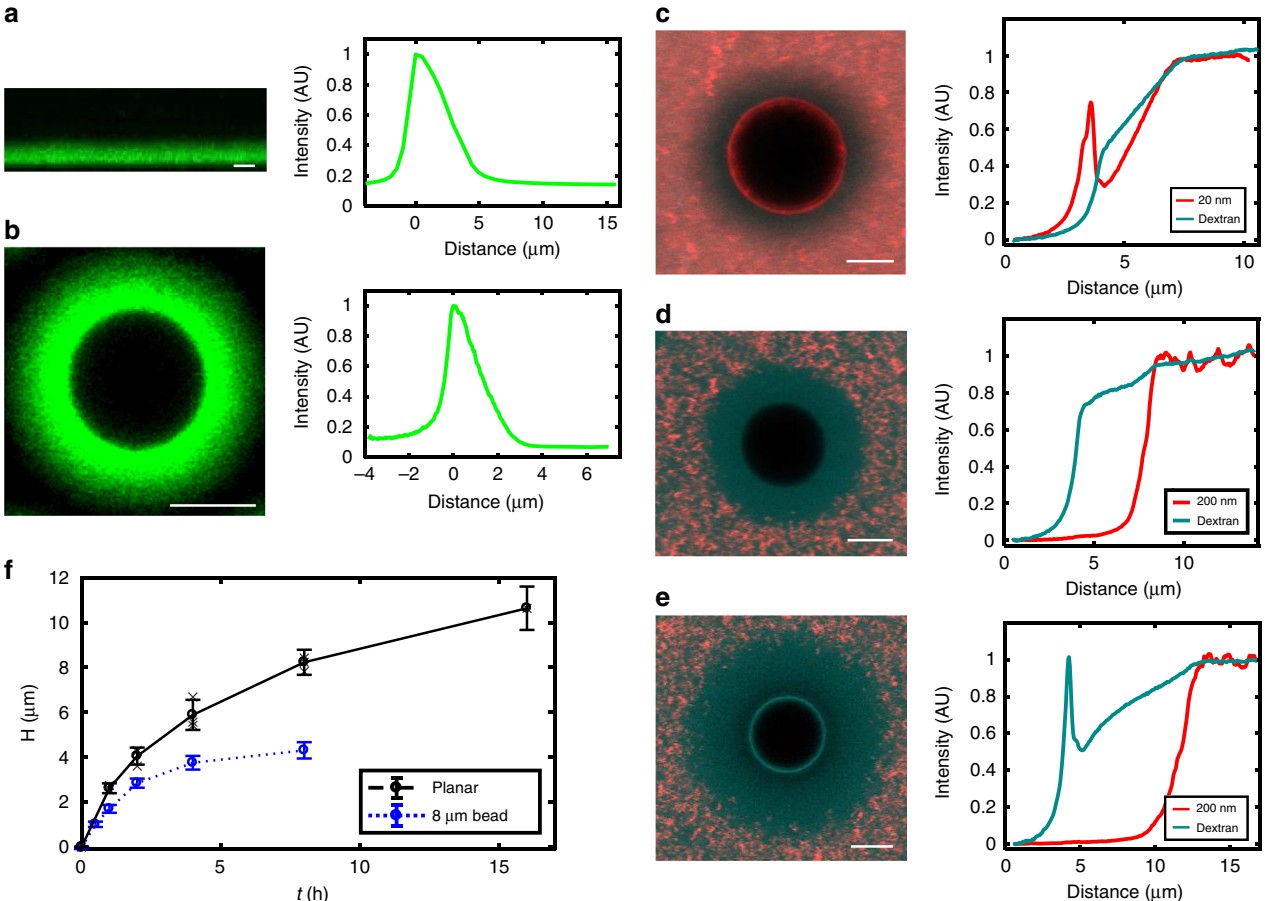

**Fig. 2** Brush concentration profiles, nanoparticle sieving and growth dynamics. **a** Fluorescent profile of a planar HA brush and its intensity profile.
**b** Fluorescent profile of spherical brush and its intensity profile. In both **a**, **b** the brush growth time is 4 h at 30 °C and the ionic strength 150 mM. **c**−**e**
Cross-section of brush grown for 4 h on 8 μm microsphere (left) and the particle penetration intensity profile (right) for **c** 20 nm, 150 mM, **d** 200 nm,
150 mM and **e** 200 nm, 1.5 mM. Twenty nanometer particles show a distinct gradient within the brush while 200 nm particles remain excluded. Dextran
(10 kDa, ~5 nm) is present in all cases (cyan) and shows an enhanced gradient in the low ionic strength swollen brush. **f** Dynamic growth of hyaluronan
brushes generated by HA synthase (150 mM). The brush height reaches $2.62 \pm 0.2$ μm (st. dev.) in just 1 h (planar) in high ionic strength conditions. After
16 h the brush is $10.6 \pm 1.0$ μm (st. dev.). Measurements from individual brushes are shown in gray, averages and standard deviations are indicated in black.
$N\_planar = 3$ brushes, 12 regions per sample, except 16 h brush which is just one brush. The spherical brush plateaus at much earlier times (~5 h) at a final
height of $4.3 \pm 0.4$ μm (st. dev.). $N\_spherical > 120$ for spherical brush height measurements. All scale bars are 5 μm.

GA/membrane fragments and HA brush, we were able to show
that increasing HA deposition leads to systematic changes in the
spectra consistent with making them more like pure HA films
(see Supplementary Note 2, Supplementary Table 2, and
Supplementary Fig. 5).

The dynamic generation of HA brushes by HA synthase
(Fig. 2f) was characterized next. Measurements indicate that the
brush height ranges from a few microns at early times up to 10.6
± 1.0 μm after 16 h in physiological conditions (high ionic
strength, 130 mM, ± is st. dev.) on planar surfaces (Supplemen-
tary Note 3 and Supplementary Fig. 6). On the microspheres
(always 8 μm diameter in this manuscript), the maximal spherical
brush height is significantly lower, plateauing at 4.3 ± 0.4 μm after
8 h (130 mM, ± is st. dev.). The height difference between the
planar and spherical surfaces likely arises from geometrical
effects, which on positively curved surfaces like spheres leads to
more accessible free volume as the polymers extend radially[47].
The HA brush height was determined using particle exclusion
assays (Figs. 1h, 2c–e), a common approach to characterizing
HA-rich glycocalyx on cells[48–50]. Fluorescently labeled polystyr-
ene nanoparticles >100 nm stop roughly at the interface of the

brush (Supplementary Fig. 7). Comparison of estimates of brush
thickness by fluorescence labeling and 200 nm particles showed
statistically equivalent results.

HA synthase-generated HA films thicker than $H \sim 300$ nm
should lie well within the brush regime, which is defined by the
requirement that the average distance between grafting points is
less than the diameter of the polymer in solution. The
hydrodynamic radius for HA is $R_H \sim M_W{}^\nu$, where $0.6 < \nu < 0.8$
for high MW HA[51]. Using $\nu \sim 0.7$, we find that at 1 h, $R_H$
~ 324 nm. Since the brush height increases linearly in early times,
at 10 min the brush height should be approximately ~333 nm.
Correspondingly, the HA $M_W$ increases linearly at early times as
well[31], yielding 0.72 MDa or $R_H \sim 100$ nm. The grafting density is
740 chains per μm² or 1 chain every 37 nm, much less than a
chain diameter of 200 nm. Hence at 10 min, with a 300 nm brush,
the typical chain will have a diameter five times the average
distance between chains, putting the system well into the brush
regime.

Examination of the size-dependent penetration of the brushes
by nanoparticles reveals that under physiological conditions,
200 nm and larger diameter particles are repelled from entry into

the brush, while 100 nm and smaller penetrate to varying extents (Fig. 2c–e, Supplementary Fig. 7). A concentration gradient throughout the spherical brush is clearly visible for the 20 nm particles (Fig. 2c, $t_{growth} = 4$ h, 130 mM). The 20 nm particles penetrate the brush, with approximately 30% of the bulk concentration value reaching the surface. Fluorescent negatively charged dextran (~10 kDa, ~5 nm diameter[52]) shows a 30–50% approximately linear decrease in intensity in the presence of 200 and 20 nm particles at high ionic strength. When exposed to highly extended brushes at low ionic strength (~1 mM), the dextran gradient is pronounced enough to be visible by eye and drops by 50% (Fig. 2e). The observed gradient may arise from spatially varying density of dextran arising from excluded volume and/or charge effects; further the brush may act as a sieve to size separate the polydisperse dextran. While traditional dense synthetic brushes exclude molecules[53], the partial penetration into these spherical HA brushes is not unexpected given the low grafting densities, the spherical geometry, and the broad polydispersity—all of which are expected to enhance particle penetration into brushes[54]. These studies show the potential to use HA brushes to sieve objects and molecules.

Hyaluronan is a weak polyelectrolyte and therefore sensitive to ionic strength and changes in pH. Figure 3 illustrates the stimulus responsiveness and reversibility of the HA brushes to changes in ionic strength. The planar brushes nearly double in height upon reduction of the ionic strength from 130 to 1.3 mM. For brushes grown for 16 h, the height increases from $12.1 \pm 0.2$ to $22.2 \pm 2.5$ μm (± is st. dev.), and it can be reversibly collapsed and stretched by exchange of the solvent, as shown in Fig. 3. Similar results were found for brushes grown for 4 h which increase more than 200% from ~7 to ~15 μm (see Supplementary Fig. 8). In both cases, there is a small decrease in height with each solvent exchange, with more HA loss from the inherently thicker brushes. We studied the impact of brush loss from pipetting effects on a 5.5 h growth brush, and found that the height change is consistent with loss from solvent exchange (shear-induced desorption), rather than the natural desorption of HA from HA synthase over short time periods (Supplementary Note 4 and Supplementary Fig. 9).

**Brush regeneration and on-demand synthesis.** The HA synthase maintains the ability to generate multiple polymers sequentially after natural release or external degradation of the HA[31]. Hence, the HA synthase interfaces should be able to regenerate or continuously replenish an HA reservoir as needed. To demonstrate this, we grew the brush for 1 h on planar interfaces, removed it by enzymatic degradation (hyaluronidase) and then regrew it three times—each time, after enzymatic removal (Fig. 4a, b). Little deviation in final brush height from the initial height is detectable with each subsequent regeneration. The small decay in brush height after each regeneration step may be due to the hyaluronidase sticking to the surface. We found that we had to minimize hyaluronidase adsorption to the surface using bovine serum albumin (BSA), or no brush would grow. It is possible that small amounts of hyaluronidase still bind to the surface, increasing with each exposure. These experiments demonstrate the potential for HA synthase brushes to be employed as regenerative interfaces.

The system also has capability for on-demand synthesis. Figure 4c shows how the HA polymerization and brush growth can be halted at a desirable height and then later, reinitiated by controlling the availability of $Mg^{2+}$ ions and sugar substrates. In this experiment, the brush growth was halted after 2 h, paused for 30 min and then reinitiated for 2 h. Despite the pause, the final brush height was in the expected range of brush heights after 4 h uninterrupted growth.

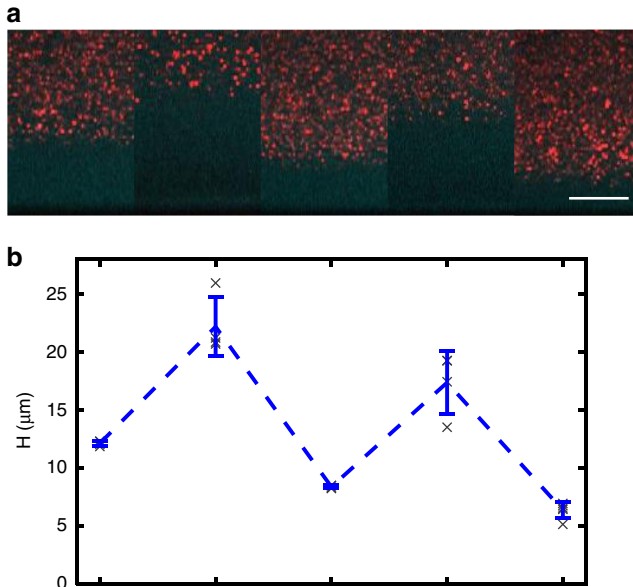

**Fig. 3** Stimulus responsiveness and reversibility. **a** Brush height during a series of solvent swaps from 133 to 1.33 mM for a brush previously grown for 16 h. Scale bar, 10 μm. **b** Quantification of the brush height shows that at ultra-low ionic strengths, the brush stretches out by nearly 200%, peaking at $22.0 \pm 2.5$ μm (st. dev.) during the first exchange. While the brush swelling and shrinking is reversible, the repeated handling (and tension induced by stretching) leads to some loss of the HA, which is weakly bound to the HA synthase. As a consequence, a gradual decrease in the overall brush height is observed. Each gray data point corresponds to five independent measurements ($211 \times 211$ μm² area) from one sample. Blue data points show the mean and st. dev.

**Brush engineering: stabilization and micropatterning.** Some applications like lubrication or antifouling interfaces or fundamental polymer physics studies will require a permanently stabilized brush. This is not possible with HA synthase binding alone, as the polymers release from the enzymes over a few days (Fig. 4d). Thus, we developed a strategy to generate stabilized brushes by crosslinking the HA to the underlying PEI surface chemistry with EDC-NHS chemistry. The EDC activates the carboxyl groups on HA for conjugation to secondary amine groups available from the PEI of the underlying grafting surface.

The surface-reinforced brushes are slightly reduced in height after processing, but remain stable for up to a year with weak decay in thickness over long times (Fig. 5a, Supplementary Fig. 10). Reinforced spherical brushes have similar exponential-like decay concentration profiles as untreated spherical brushes (Supplementary Fig. 11) and when exposed to hyaluronidase the brush is quickly eliminated (Supplementary Fig. 12). For unreinforced HA brushes, exposure to SDS quickly eliminates the HA structure by destroying the membrane fragments (Fig. 5b). For reinforced HA brushes, SDS does not eliminate the brush, confirming that the processing achieves covalent linkage of HA to the underlying surface (PEI) rather than the membrane fragments or the HA to itself (Fig. 5c).

Patterning polymer brushes is an important component of brush engineering for applications[55]. We therefore established a method to photopattern the HA brushes by deactivating HA synthase function with UV irradiation. Using a confocal microscope equipped with a 405 nm laser, we found that energy densities above 2.13E-04 W s per μm² are sufficient to destroy HA synthase enzyme activity and prevent HA growth in

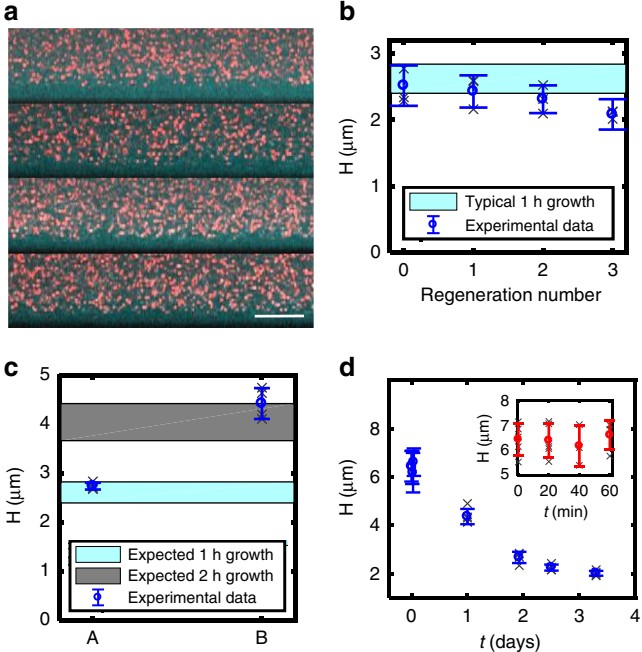

**Fig. 4 Brush regeneration and patterning. a** Regeneration of HA brush after enzymatic degradation with hyaluronidase. Top image shows brush after 1 h of growth before digestion. The next three images show the regenerated brush following digestion and 1 h regrowth 1, 2, and 3 times. **b** Brush height versus the number of regeneration times ($N = 3$ brushes, where gray $x$'s correspond to average of five measurements of each brush and blue is the mean and st dev). **c** Interrupted growth (A) followed by an additional growth period of 1 h (B). (Gray $x$'s correspond to five measurements from one sample.) **d** Brush stability versus time (unreinforced, natural brush), For both **c, d**, $N = 1$ brush, gray $x$'s correspond to measurements on same brush, blue reports the mean and st. dev.

micropatterned regions (Fig. 5d, e). Figure 5e shows a checkerboard pattern of the brush comprised regions with active HA synthase and other regions with no active HA synthase ($t_{growth} = $ 4 h, 130 mM). This simple, scalable method enables the binary patterning of HA polymer brushes, and could be extended to mask aligners to rapidly pattern larger areas. Such engineered interfaces should have applications in various fields including controlling cell adhesion, tuning wettability, and indirectly modifying the height of brushes[56]. Further, future work focused on tuning the power density delivered to the HA synthase interfaces may enable a simple approach to controlling its density, and consequently the brush grafting density.

**Cell-brush and biofilm interactions for biomaterials applications.** Motivated by the possibility of biomaterials applications such as implants and treatment of chronic wounds[57], we explored the interaction of mammalian cells and bacteria with the HA brushes. Mouse embryonic fibroblasts (MEF) seeded on top of reinforced HA synthase brushes accessed the underlying glass substrate within 1–3 h ($t_{growth} = 8$ h; see Movie 1, Fig. 6a–e). Particle exclusion assays reveal that the MEFs express their own extensive HA-rich glycocalyx[58] around the cell by 12 h (Fig. 6d, e). Once in contact with the substrate, the fibroblasts rapidly adhered, displaying their normal phenotype (see Supplementary Fig. 13a), and continued to proliferate for 12 h when imaging was stopped. A cell viability assay showed that of ~3000 measured cells on reinforced brushes, only six were dead (see Supplementary Fig. 14).

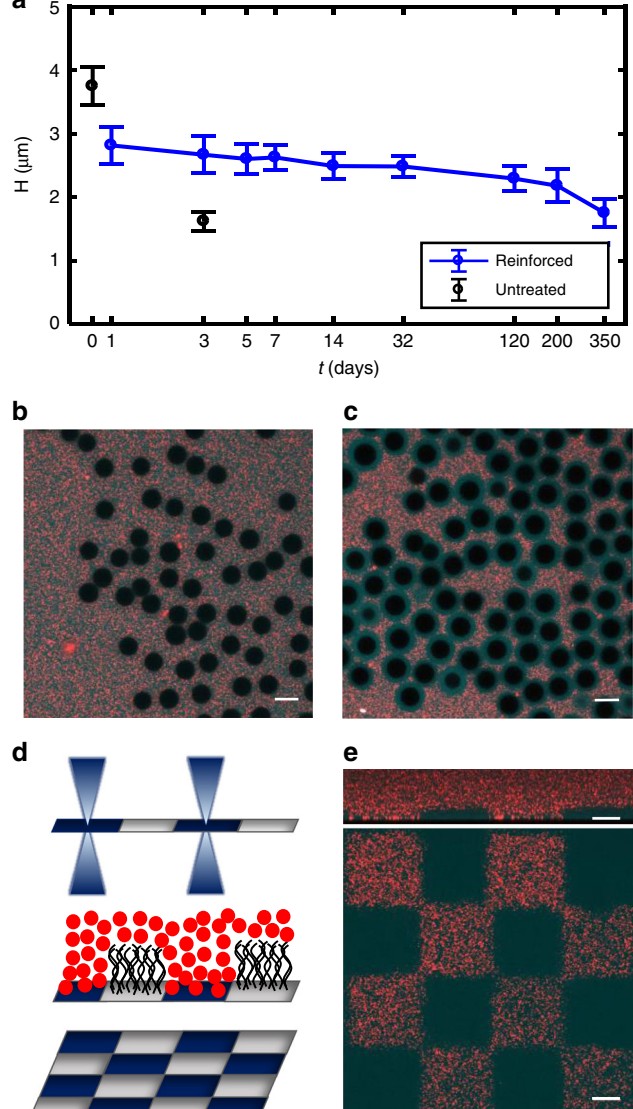

**Fig. 5 Reinforcement stabilizes HA brush. a** Reinforced spherical brush height measured over a period of 1 year versus natural, untreated spherical brush height. For each time point the corresponding number of measurements are $N = 35, 126, 119, 103, 64, 56, 62, 76, 59, 28$. Error bars report st. dev. **b** Reinforced brushes on spherical particles are still digested by hyaluronidase, indicating minimal HA−HA crosslinking within the brush. **c** Reinforced brushes on spherical particles resist detergent treatment (SDS) despite the removal of membrane fragments. This confirms the HA is stably bound to the underlying glass substrate. **d** UV micropatterning of HA synthase activity with a confocal microscope ($\lambda = 405$ nm). **e** Checkerboard pattern illustrating binary patterning of the brush. Red regions indicate areas where no brush grew; cyan regions indicate regions where brush expels red nanoparticles. Top: $XZ$ confocal side view of micropatterned brush. Bottom: $XY$ confocal image taken at the glass interface. All data in this figure were acquired in physiological conditions (150 mM). All scale bars are 10 μm.

Fluorescent labeling of the brush after 12 h of cell interaction showed visible black tracks at the surface, consistent with partial digestion of the brush (Fig. 6a). The spatial range of the brush digestion was relatively localized, consistent with the activity of endogenous membrane-bound hyaluronidase in MEF cells[59]. Furthermore, particle exclusion assays of the brush-cell samples revealed that the brush had become unusually inhomogeneous in

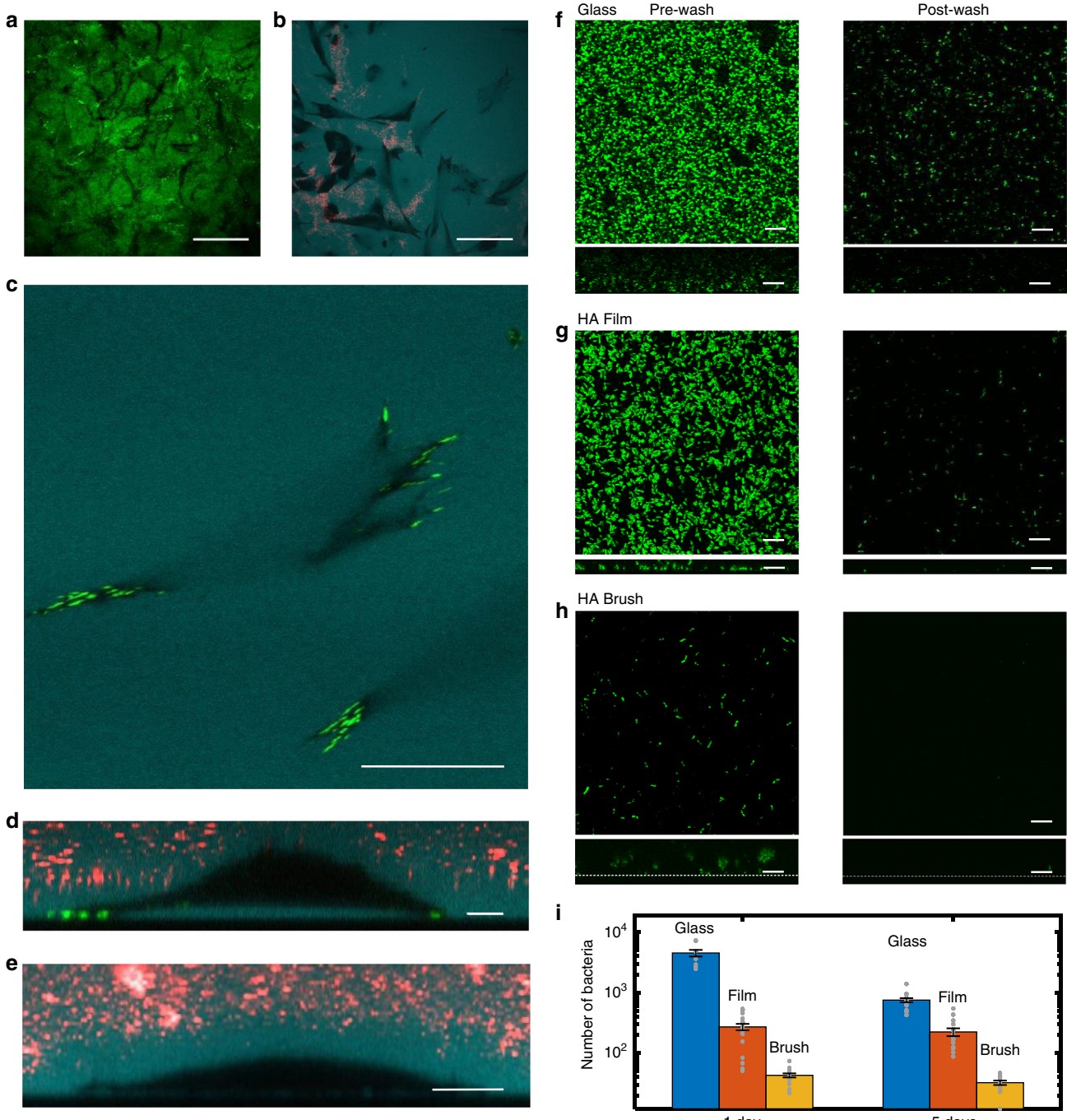

**Fig. 6** Fibroblast and biofilm interactions with HA brush. **a** Black tracks in the GFPn-labeled HA brush reveal areas of brush digestion by MEF cells. Scale bar 50 μm. **b** Fluorescent dextran (cyan) highlights space under adherent cells (to the brush). Black regions correspond to cell area in close contact with surface. Red fluorescent particles fill areas of brush elimination due to digestion by cells. Scale bar 50 μm. **c** Fibroblasts attach to the interface underlying the HA brush, expressing mature focal adhesions (green, vinculin). Dextran is apparent underneath the spread cell. Scale bar 20 μm. **d** z-slice of the same adherent fibroblast from (**c**) sitting on a cushion of HA. Scale bar 10 μm. **e** Particle exclusion assay reveals MEFs express a thick glycocalyx. All MEF images were taken after 12 h exposure to the HA reinforced brush. Scale bar 10 μm. **f−h** Confocal micrographs of GFP-producing *Pseudomonas aeruginosa* (PAO1) interacting with a glass interface (**f**), an HA film (**g**), and a reinforced HA brush (**h**). All images were taken at the glass interface. Left: biofilm growth before washing (1 day). Right: biofilm growth after washing (1 day). Dextran was used to identify the glass interface beneath the brush. *XZ* side views of the biofilms are presented below each respective *XY* top view of the samples. Scale bars, 10 μm in (**f−h**). **i** Comparison of the number of bacteria retained after washing different surfaces. Data were taken in triplicates and averaged over five regions per sample. Error is SEM. Data are available in Table 3 in the Supplementary Information.

height. When focusing at the surface of the sample (defined here as the base of the brush), no beads should be visible if the brush remains intact (outside of the cell areas). Yet frequently, the beads were clearly visible at the surface, indicating that the brush had been compromised or removed (Fig. 6b and Supplementary Fig. 15 for xz confocal images). The destruction of the brush must be cell driven, since the reinforced brushes are homogenous and stable over months.

Previous studies have shown that in order for cells to adhere to high molecular weight HA films, membrane-bound hyaluronidase removes the HA film under the cell[60,61]. Indeed, in the areas around the focal adhesions, we found fluorescent dextran (10 kDa) was excluded from under the cell, indicating the very confined distance between the cell and the surface. Yet, despite the presence of focal adhesions and localized HA removal, well-spread MEFs frequently maintained a significant HA-rich gap with the substrate ($2.7 \pm 1.5\,\mu m$, $N = 21$ at maximum separation, see Fig. 6c, d, $\pm$ is st. dev.). The voluminous space beneath the cells is evident when examining fluorescent dextran (10 kDa) penetration under the cell, which is only present for cells adherent to HA brush interfaces (Fig. 6c, d). The same cells when bound to glass (or to membrane fragment-coated glass with no HA growth) display no measurable gap and no penetration of dextran under the cell (see Supplementary Fig. 13c). The variability in the dextran intensity (cyan) the cell−brush interface (see Fig. 6b, c and Supplementary Fig. 13c) is consistent with the interpretation that some regions beneath the cell have more or less digested HA.

How this final cell−brush configuration is realized is unclear. Without some method to compress or digest the underlying brush, the cells could not reach the underlying surface. Gravitational force plays a minimal role. For example, 8 μm silica spheres remain suspended above the brush with little brush deformation (Supplementary Fig. S16). An (inactive) cell of lesser density ($\rho_{cell} \sim 1.1\,g$ per $cm^3 < \rho_{silica} \sim 2.65\,g$ per $cm^3$) but of slightly larger size would also remain suspended. Hence, two possible mechanisms could enable cell penetration of the brush and adhesion to the surface. The first is membrane-bound hyaluronidase digestion of the brush, a hypothesis for which some supporting evidence has been provided. Second, the cells may be able to manipulate the brush through adhesive interactions via HA-binding proteins like CD44, and/or extend protrusions through the HA brush to reach the surface and then exert contractile forces to compress the brushes as the cells approach the interface.

In summary, MEF cells flourish within the brush environment. These preliminary findings demonstrate that the HA brushes will be a useful platform to systematically study the interplay of sugar-rich hydrogels and glycocalyx in cell adhesion, an area of growing relevance and interest[62–66].

The prevention of bacterial infection is another crucial aspect of both implants and wound treatments—in particular the problem of biofilm formation and adhesion. Encouraged by the observation that the brushes sterically exclude particles greater than 100 nm, and that HA is a known antifouling polymer[67–70], we examined whether the biofilm forming bacteria *Pseudomonas aeruginosa* (PAO1) are capable of irreversibly attaching to the brush interface and establishing a biofilm. Both natural and reinforced brushes repelled the bacteria, natural brushes for at least 1 week (until the brush decays) and reinforced brushes for at least 2 weeks when the experiments ended—as determined by high-resolution confocal microscopy. Few bacteria reached the underlying substrate in this time. In both cases, removing the bacteria from the brush interfaces was straightforward, and verified with confocal microscopy.

Next, systematic characterization of the brush-bacterial interaction was repeated three times, exposing reinforced brushes to GFP-expressing PAO1 bacteria for 1 and/or 5 days, rinsing, and then imaging remnant bacteria attached to or embedded within the brush. Each sample was randomly imaged at five locations and the total volume occupied by bacteria assessed to estimate the number of bacteria (Fig. 6f–i). The results were compared with PAO1 biofilm formation on glass and HA thin films. The images in the left column correspond to the three surfaces (glass, HA film, HA brush) prior to washing. The right column shows the same samples after washing. Quantification of the total number of bacteria under each condition is summarized in Fig. 6i (see Supplementary Table 3). The percentage of bacteria relative to the glass slide was dramatically decreased on both the HA film and even more so on the HA brush at 1 and 5 days (Fig. 6i, and Supplementary Table 4). The glass surfaces retain 99% more bacteria than the HA brushes after 1 day. As expected, the HA film shows improved resistance to fouling, yet it still retains six times more bacteria than the HA brush (Fig. 6h) after both 1- and 5-day exposures. On the HA brushes, only 1–2% of the remnant adhered bacteria were found at the underlying brush−glass interface. Addressing the brush uniformity, not yet optimized, could further improve the outcomes.

In future work, the antifouling properties of the brush could be complemented with doping the brush with antimicrobial compounds using straightforward and accessible chemistry associated with HA. Further, the selective biointeractivity, i.e. inhibition of pathogenic microbial adhesion but enhancement of beneficial host cell responses, establishes the HA brush interface an interesting option for coating implants, bandages or other materials with similar requirements[71].

## Discussion

The enzyme-generated giant polymer brush introduced here is distinct from traditional polymer brushes in several ways. Notably, rather than grafting-from or grafting-to the surface, the HA brush is generated by a transmembrane protein that extrudes the HA through a protein pore. The enzyme is able to polymerize much higher molecular weight molecules than most other techniques such as free radical polymerization with rare exception[72,73]. Further, the polymer grows from the base of the interface rather than the tip of the polymer. The extraordinary brush height results in the unusual ability to directly visualize the brush with detailed spatial resolution. Last, but importantly, the enzymes enable the option of brush regeneration.

We consider factors that may affect future applications. Biomedical tools that could benefit from this technology such as implants or bandages may require large surface areas. It is possible to scale up the surface area covered. We estimate that 1 L of (*E. coli*) bacterial culture can be used to cover ~2 m² of area. Glass surfaces prepared with HA synthase fragments can be stored for up to at least 9 months at −80 °C with similar performance (spherical brushes made from fresh surfaces have $3.8 \pm 0.3\,\mu m$, $N = 146$, $t_{growth} = 4\,h$, $N = 146$; 9-month-old have $3.9 \pm 0.2\,\mu m$, $N = 32$, $\pm$ is st. dev.). Indeed, longer functionality is expected since the fragments can be stored for at least 12 years and maintain similar functionality (spherical brushes made with these fragments have brush heights of $3.0 \pm 0.3\,\mu m$, $N = 147$, $\pm$ is st. dev.). Binding the membrane fragments to other materials such as plastic or titanium, i.e. for catheters or implants, should be achievable using modified surface functionalization schemes.

Two concerns arise with the system in the context of biomaterials applications and regeneration. The first is that the bacterial membranes likely contain lipopolysaccharide (LPS), which could stimulate an immune response. One approach to bypass the LPS trigger is to work with endotoxin-free bacteria that are nonimmunogenic[74,75]. Another strategy is to use genetically

engineered mammalian cells as the source of HA synthase, possibly even from the future host of the biomaterial. The second concern is the stability of the HA synthase and its ability to continually synthesize HA. HA synthase activity decays over time. This timescale would limit the window of opportunity for regeneration of interfaces or use of an interface to continuously release HA while maintaining a protective HA brush. If there is sufficient interest, protein engineering may enable the development of a HA synthase that is stable for longer periods of time.

The HA brush platform also presents a myriad of opportunities for fundamental polymer physics studies, where many open questions remain in the arena of polyelectrolyte brushes[76,77]. The system enables the direct visualization of the detailed spatial structure of brushes and ultimately, possibly even dynamics of individual strands. For example, both polymer concentration profiles with height and particle penetration or protein absorption can be studied directly and compared with theory[78]. Schemes to fluorescently label only the tip of the brush as it extrudes, by for example stopping after small growth, and then continuing growth after labeling, will facilitate dynamic studies of individual polymer strands and/or the ends of the polymers[79]. In another example, the curvature-dependent brush height data shown in Fig. 2f deserves further systematic study and comparison with theoretical predictions[47]. Studies like these, combined with full characterization of the polymer length distribution similar to that shown in Fig. 1f will be a very exciting and powerful tool to make progress in polymer physics.

Engineered biomaterials based on HA are abundant in the literature and medical practice. The regenerative HA brush interface introduced here represents a platform for creating a distinct class of interfacial HA biomaterials[80]. The straightforward and established protocols to chemically modify and cross-link HA present numerous opportunities for exploration and expansion of traditional approaches[22,25]. The regenerative ability of the platform is a useful tool to explore, as are the distinct properties of a gigantically thick brush[81]. From lubrication to antifouling to molecular filtration, the unusual characteristics of this brush motivate a plethora of tangible research directions. In addition to these obviously immediate applications, the brush will enable fundamental studies of polyelectrolyte brushes as well as the cell glycocalyx, which plays an important but poorly understood biophysical role in the body[65].

## Methods

**Dynamic light scattering**. The membrane fragment suspension was removed from the −80 °C freezer and defrosted. Then, it was diluted from 1 mg/mL to a final concentration of 0.02 mg/mL with phosphate buffer (pH 7.3, 75 mM NaKPO$_4$, Na$_2$HPO$_4$ (J.T. Baker 3827-01), KH$_2$PO$_4$ (Sigma-Aldrich P5655)) together with 0.1 mM EDTA (Mallinckrodt Chemicals 2590-12), 50 mM NaCl, 2% glycerol, 5 mM DTT (Sigma-Aldrich D9779). Dynamic light scattering was performed and analyzed using a Malvern ZetaSizer Nano ZS. The autocorrelation function was acquired every 10 s with ten measurements for each of three runs. The intensity of light was collected at a scattering angle of 90°. Analysis yielded an average membrane fragment diameter of ~145 nm (before HA growth).

**Activating HA synthesis**. The storage buffer was exchanged with activation buffer (pH 7.3, 75 mM NaKPO$_4$, 50 mM NaCl, 20 mM MgCl$_2$, 0.1 mM EDTA). After warming the sample for 45 min in a 30 °C incubator, uridine 5-diphosphoglucuronic acid trisodium salt (UDP-GlcUA, Sigma-Aldrich U6751) and uridine 5-diphospho-*N*-acetylglucosamine sodium salt (UDP-GlcNAc, Sigma-Aldrich U4375) were added to a final concentration of 5 mM.

**Quenching HA synthesis**. The activation buffer was thoroughly exchanged with quenching buffer (pH 7.3, 75 mM NaKPO$_4$, 50 mM NaCl, 20 mM EDTA) via pipetting and mixed by gently pipetting the liquid up and down a few times (repeat buffer exchange seven times).

**HA purification for solid-state nanopore analysis**. HA grown from bacterial membrane fragments in suspension was purified for later quantification by solid-state nanopore. After HA production, EDTA was added to a final concentration of 40 mM to quench the growth and the solution was put on ice for 10 min. The solution was then placed on a 90 °C heat block for 10 min and subsequently put on again for 1–2 min to inactive the HA synthase[82]. To dissociate HA from the synthase, the solution was mixed in a 3:1 ratio of Folch to HA solution and allowed to shake for 15 min. The Folch/HA solution was centrifuged for 7 min at 7168 × g (8000 rpm) and then the supernatant containing HA was removed and speed vacuumed until the solution volume was reduced to one-third.

**Solid-state nanopore determination of HA molecular weight distribution**. Enzymatically generated HA samples were mixed with measurement buffer (6 M LiCl, 10 mM Tris, 1 mM EDTA, pH 8.0) to a final concentration of 30 ng/μL and stored at −20 °C until measurement. Solid-state nanopore analysis was performed on 10 μL aliquots of the samples. A single pore (6–8 nm diameter) was fabricated[83] in a 19 nm thin, free standing silicon nitride membrane supported by a silicon chip (4 mm) and was placed in between one reservoir of clean measurement buffer and one reservoir of sample mixture. Ag/AgCl electrodes (Sigma-Aldrich, St. Louis MO) were placed in each reservoir and an Axopatch 200b patch clamp amplifier (Axon Instruments, Union City, CA) was used to both apply a voltage of 200 mV and record trans-pore ionic current and resistive pulses caused by HA translocation through the pore. Data were collected at a rate of 200 kHz with a four-pole Bessel filter and an additional 5 kHz low-pass filter was applied using custom software. Resistive pulses ("events") in the current signal were identified as transient interruptions in the ionic current >5σ in amplitude from the baseline and with a time duration range of 25 μs–2.5 s. The event charge deficit (ECD[84]) (defined as the integrated area of the event) was determined for each translocation event and converted to molecular weight using a calibration standard produced with synthetic, quasi-monodisperse HA[37]. An MW distribution histogram was generated for each sample with these values and used for subsequent analyses.

**Immobilizing bacterial membrane fragments on glass slides**. Coverslips (VWR 48366 246 or VWR 48366 067) were sonicated in ultrapure water for 15 min and cleaned in reagent grade acetone in a sonicating water bath for 15 min. The coverslips were then rinsed with ultrapure water and dried with nitrogen and treated in a plasma cleaner (Harrick Plasma, PDC-32G, High RF power, air, 1 min). Poly (ethyleneimine) (PEI) (Sigma 482595, average $M_w$ 1.3 kDa, 50% w/v in H$_2$O) was diluted with ultrapure water to 2.5% and the pH was adjusted to 7.0 using HCl. Two hundred microliters PEI was used to cover each coverslip (facing up) after plasma cleaning. The coverslips were incubated for 1 h before they were rinsed with ultrapure water and dried with nitrogen. Glutaraldehyde (Sigma G7651, average $M_w$ 0.1 kDa, 50% w/v in H$_2$O) was diluted to 2.5% with PBS. Two hundred microliters glutaraldehyde was sandwiched between a piece of parafilm and the coverslip, with the PEI-coated side facing the solution. The coverslips were incubated for 1 h before they were rinsed with ultrapure water and dried with nitrogen. The coverslips were then mounted on custom Teflon rings using vacuum grease to seal. Thirty microliters of 0.2 mg/mL HA synthase-rich bacterial membrane fragments (diluted from 1 mg/mL in phosphate buffer) was pipetted into each teflon ring. The coverslips were incubated for 1 h. The solution in the sample holder was exchanged four times with Tris storage buffer (pH 7.3, 50 mM Tris (BDH 0312), 500 mM NaCl, 20 mM DTT, 5% glycerol). The samples were stored at −20 °C.

**Immobilizing bacteria membrane fragments on silica microspheres**. In a 90 °C water bath on a stirring hot plate, 100 mg monodisperse silica beads (Cospheric LLC, 1.8 g/cm$^3$, 7.75 μm diameter, CV = 3.7%, <1% doublets) were added to 1 mL 30% H$_2$O$_2$ in a 1.5 mL centrifuge tube. The beads were briefly vortexed to suspend them and then the tube was put in a sonicating water bath for 15 min to disperse the beads. A stir bar was added along with 3 mL sulfuric acid (BDH Aristar Plus, average $M_w$ 0.98 kDa) in a disposable glass vial (22 mL, VWR 470206-384). The 1 mL beads suspended in 30% H$_2$O$_2$ were gently pipetted into the vial while allowing slow stirring. The vial was secured with a clamp in the stirring hot water bath. The piranha cleaning was allowed to proceed for 2 h during which the level of the water bath is maintained. Finally, 10 mL ultrapure water was slowly added to the vial, allowing the water to fully mix. Heating and stirring was turned off and the vial was allowed to sit overnight to let the microspheres to settle. On the next day, most of the piranha solution was removed using a glass pipette tube. The beads were resuspended with stirring and then transferred to a glass centrifuge tube. Using a swing bucket centrifuge, the solution was exchanged with ultrapure water seven times and checked to ensure that the pH of the solution was above six. After supernatant removal, the beads were suspended in 2.5% w/v PEI and then stirred for 1 h before washing seven times with ultrapure water. Glutaraldehyde was added to a final concentration of 2.5% w/v with PBS to a clean glass vial. The washed beads were slowly added and the suspension was allowed to stir for 1 h before washing seven times with ultrapure water. To prepare the membrane fragments for deposition on glutaraldehyde-modified beads, exchange the Tris-based membrane fragment storage buffer into a phosphate buffer using a Slide-A-Lyzer™ MINI Dialysis Device (ThermoScientific 69570) in an ice box (4 °C). The phosphate buffer was refreshed twice in 45 min intervals. The fragments were transferred to a 1.5 mL centrifuge tube and the activated silica beads were slowly added. The centrifuge tubes were placed in a tube rotator and set to slow rotation for 2 h at

4 °C. The beads were allowed to settle and then the supernatant was removed. The beads were exchanged into the Tris storage buffer by adding the buffer, allowing the beads to settle, and removing the supernatant three times. Finally, the beads were aliquoted in storage buffer and stored in −80 °C freezer.

**Confocal imaging**. Fluorescent characterization of the HA brush was made using a scanning laser confocal microscope (FV1000, Olympus, Tokyo, Japan; Objective: PlanApo N, ×60/1.42 NA oil). When imaging planar polymer brushes, a 100 nm vertical step was used and a 20-μm-thick z-stack was taken of the planar brush. When imaging the polymer brushes on microspheres, a 30−60 nm horizontal pixel size and 470 nm vertical step was used. A 4-μm-thick z-stack was taken for each microsphere. The vertical range was selected to, at minimum, measure to slightly above the bead center. Imaging was completed within 1 h after quenching HA synthesis in order to avoid significant desorption of the HA polymers.

**HA brush concentration profile analysis**. The concentration profile of the HA brush is estimated by looking at the fluorescent intensity generated by GFPn, a protein that binds to HA. The GPFn is comprised of green fluorescent protein (GFP) with the HA-linking domain of neurocan connection, specifically binds to HA[45]. GFPm was expressed by HEK 293 EBNA cells purified according published protocols[46,50]. In experiments where HA-GFPn profiles were assayed, we incubated with 8 mg/mL BSA for 40 min to backfill the surface before HA synthesis. This reduces the adhesion of GFPn to the underlying substrate. Before starting the HA synthesis, the BSA solution was removed. After quenching HA synthesis, we added ~44 μM GFPn to a final concentration of 3 μM GFPn. For planar brushes, the z fluorescence profile was extracted from confocal microscopy z-stacks. For spherical brushes, the centers of the microspheres are identified by segmenting either the image of the dextran channel or the GFPn channel using Otsu's method[85]. An azimuthally averaged profile was extracted from 30° cones from the center of a microsphere, only analyzing the cones that are clear of neighboring brushes. The mean background (and its standard deviation) was determined from the last 50 data points at the edge of the image (away from the microsphere). The edge of the brush was defined where the intensity is greater than or equal to twice the standard deviation above the mean background. The peak in the GPFn was identified as the surface of the microsphere. The thickness of the brush measured using 200 nm particle exclusion assays is $3.75 \pm 0.30$ μm ($N_{beads} = 146$) is consistent with the value measured using GPFn of $3.70 \pm 0.25$ μm ($N_{beads} = 112$) for nonreinforced brushes on 8 μm diameter particles ($t_{growth} = 4$ h, high salt condition).

**Particle exclusion and nanoparticle penetration assays**. Particle exclusion assays were performed by adding red 200 nm FluoSpheres (carboxylate-modified Molecular Probes, Inc., catalog number: F8810) to a final concentration of 0.7% w/v along with fluorescent dextran (Molecular Probes, Inc. Alexa Fluor 647, 10 kDa) to a final concentration of 33 μg/mL. Nanoparticle penetration was investigated using the red, 200 nm FluoSpheres, green 100 nm Fluospheres (Catalog number: F8803), and green 20 nm Fluospheres (Catalog number: F8787). When the grafting surface needed to be labeled, 0.007% w/v of the green, 20 nm nanoparticles were added. For details of planar brush height analysis using 200 nm particles, see Supplementary Note 3 and Supplementary Fig. 6.

**Stimulus response and reversibility to ionic strength**. The sample was first imaged to establish a baseline height for the brush (using 200 nm particles) in the standard imaging buffer (40 μL activation buffer and 14 μL quenching buffer), which has an estimated total ionic strength of ~130 mM. The media was then exchanged three times with a 1% activation buffer diluted with ultrapure water, removing all liquid with each final wash. To image again, the 200 nm FluoSpheres were added to the sample, along with the typical volumes of activation buffer (40 μL) and quenching buffer (14 μL), but at 1% concentration diluted with ultrapure water to make the ionic strength 1.3 mM. To switch back to the normal imaging solution (130 mM), the media was exchanged three times with a 100% imaging buffer, undiluted. This process was repeated once more to study the reversibility of the brush height vs. time.

**HA brush decay assay**. To measure the short-term decay of planar brushes: Post HA growth, the synthesis was quenched and the sample was imaged using the methods described previously in the presence of a stage-top incubator to maintain the temperature at 30 °C (130 mM, $t_{growth} = 4$ h). After imaging, the sample was allowed to sit, undisturbed and then measured every 20 min. Long-term decay: for 1 year, microsphere samples were stored in the dark at room temperature in the same buffer in which they are imaged. Mineral oil was added to the surface of the buffer to hinder evaporation. Each spherical brush height measurement was performed using a fresh sample under normal imaging conditions (room temp, 150 mM).

**On-demand synthesis: pausing and restarting HA brush growth**. The sample was washed three times with a mixture of the activation buffer (but with no UDP-sugars) and the quenching buffer in a ratio of 3:1, respectively. After replacing the volume with this mixture, the sample was allowed to sit, HA growth paused, for

30 min in the 30 °C incubator. The sample was then washed with the activation buffer three times. After replacing the volume with the activation buffer, the UDP-sugars were added to a final concentration of 5 mM and growth was allowed to proceed again.

**Brush regeneration**. After exchanging the Tris storage media for activation buffer, the sample was incubated with 2% BSA (SeraCare 1900-0016) in PBS to a final concentration of 1% for 20 min in the 30 °C incubator. HA synthesis was then activated and the sample was allowed to grow for the desired time. 0.5 units/μL bovine hyaluronidase dissolved in PBS was added to the sample to a final concentration of 0.025 units/μL and allowed to sit in the 30 °C incubator for 30 min. The hyaluronidase was washed away by rinsing the sample with the activation buffer 20 times, ensuring to remove all the liquid with each wash. HA synthesis was then reactivated and allowed to proceed for the desired time. After 30 min, the hyaluronidase addition and removal (followed by a growth period of 1 h) was repeated three times. The height was measured and averaged over five areas of the sample, each $211 \times 211$ μm$^2$.

**Surface reinforcement of the HA brush**. In order to form covalent bonds between the HA polymers and the grafting surface, carbodiimide conjugation was used to crosslink the carboxyl groups on HA to the primary amine groups ($-NH_2$) on the grafting surface. At the end of HA synthesis, solution was exchanged with pH 7.0, 75 mM NaKPO$_4$, 50 mM NaCl. For this protocol, no DTT was added to the buffer because DTT reacts with EDC. Next, 100 mM EDC (1-Ethyl-3-(3-dimethylaminopropyl)carbodiimide, Sigma E1769) and 50 mM sulfo-NHS (sulfo-N-hydroxysuccinimide, Sigma-Aldrich 56485) was added to the sample. After 30 min, the solution was exchanged with newly dissolved EDC and sulfo-NHS (repeated twice). The sample was left overnight at room temperature. The next day, the solution was exchanged with a pH 8.0, 50 mM borate buffer (2 h) to quench the crosslinking reaction. Last, the reinforced brush was washed extensively with PBS.

**Bacterial fragment removal with detergent**. The detergent SDS (sodium dodecyl sulfate, Sigma-Aldrich L6026, 1 mg/mL) was used to destroy the membrane fragments, including the embedded HA synthase in order to demonstrate that reinforced brushes are bound to the underlying surface, whereas in comparison nonreinforced brushes are destroyed along with the membranes.

**Laser micropatterning of HA synthase activity**. After exchanging the Tris storage media for activation buffer, in order to define the surface, a low concentration of 8 μm silica microspheres were added to the sample and brought into focus with diffraction interference contrast (DIC) microscopy on the confocal microscope (within a stage-top temperature incubator at 30 °C incubator and a wet sponge to maintain humidity). Then, the focus was adjusted to be below the center of the microspheres an amount equal to the radius such that the focus should now be on the surface of the sample, rather than at the center of the spheres. This method allows one to focus at the plane of the membrane fragments containing HAS without introducing other chemistries or fluorescence. Then, the confocal microscope's 405 nm laser was scanned in a predetermined area using optimized settings to eliminate HA synthase activity. We found that treating the surface with an energy density greater than 2.13E-4 J/μm$^2$ was sufficient. When applying the laser to a new area, we first refocused on the surface using the microspheres as a reference. After all desired areas were laser treated, we activated HA synthesis and allowed the sample to grow for the desired time.

**Mouse embryonic fibroblast interaction with reinforced planar brushes**. Mouse embryonic fibroblasts (MEF) modified to express fluorescent (green) vinculin were used to study how cells interact with reinforced HA brushes generated by HA synthase. The cells were cultured in Dulbecco's modified Eagle's medium (D-MEM) supplemented by 10% FBS (fetal bovine serum, Corning CellGro: 35-010-CV), 1% penicillin and 4 mM L-glutamine. Brushes were grown for 8 h and then reinforced to avoid decay over the length of the 12 h experiment. The reinforced brushes were covered with cell culture media and then seeded with MEF cells. Time lapse videos were made using DIC imaging for 12 h using a ×40 Nikon Plan Fluor oil lens on a Nikon TE2000. Fluorescent images of the vinculin-rich focal adhesions were acquired using confocal microscopy after overnight seeding of the MEF cells. Vinculin-null MEFs were from Eileen Adamson (Burnham Institute, La Jolla, CA). Subsequently, vinculin-null MEFs were transduced with retrovirus to express the pXF40-eGFP-vinculin vector[86].

**HA film preparation**. Rooster comb HA (Thomas Scientific H5388, MW range $5.0 \times 10^5$–$1.2 \times 10^6$ Da) was fluorescently labeled with Alexa 546 (ThermoFisher Scientific A20002) using established protocols[87]. Briefly, the HA was dissolved in water and mixed with a 20–40-fold molar excess of dihydrazide and pH balanced to 4.75. A 4 M excess of EDC was dissolved in water and added to the HA solution and the pH was maintained at 4.75 by addition of HCl over the course of 2 h. Next, using NaOH, the pH was balanced to 7 and the HA was precipitated with chilled ethanol. To achieve 1 dye molecule per 50 nm HA, we added dye at a final concentration of 0.1 mg/mL with 2 mg/mL of the dihydrazide-modified HA. To

remove unbound dye, the solution was dialyzed with ultrapure water for 1 week, refreshing the water every day. The HA film was deposited on glass coverslips that were sonicated in ultrapure water and acetone, then plasma cleaned. The slides were then covered with 50 μg/mL of the fluorescent HA and allowed to sit in the 4 °C refrigerator overnight with wet sponges to minimize evaporation. In the morning, the slides were rinsed with DI water, dried with nitrogen, and mounted in teflon sample holders.

**_Pseudomonas aeruginosa_ bacterial culture.** A GFP-producing strain of _Pseudomonas aeruginosa_ PAO1 was streaked on a lysogeny broth (LB) (Teknova L9130) agar plate incubated at 37 °C. A single bacterial colony from the agar plate was inoculated into 5 mL of LB broth and grown overnight in a shaking incubator (3 g or 160 rpm, 37 °C). One hundred microliters of the overnight culture was inoculated into 5 mL of fresh LB broth under similar conditions until the optical density (OD) had a value of 1. This bacterial culture was then diluted to a value of OD 0.02 and used for inoculating the hyaluronan film and brush samples.

**Biofilm growth on HA brush.** Post hyaluronan growth, the synthesis was quenched and the brushes were reinforced to create stable structures. The EDC/sulfo-NHS buffer that was used to reinforce the brush overnight was replaced with borate buffer for 2 h, which was then replaced with fresh borate buffer and allowed to sit overnight at 4 °C. This was followed by two PBS buffer exchanges, each for 1 h at room temperature. One hundred microliters of the OD 0.02 culture was then added to the sample and incubated at 37 °C. For the 1-day experiments, the samples were imaged with confocal microscopy before bacteria removal. After imaging, the sample was washed in a manner where the solution was pulled up and down into the pipette three times, and for two rounds, and this procedure was repeated twice with fresh LB broth both times. The washed samples were topped with fresh LB and imaged again to check for bacterial adherence and biofilm formation. Twenty nanometers fluorescent beads were gently added and imaged to locate the coverslip surface. For the 5-day experiments, the samples were replenished with fresh LB broth every 24 h and on the fifth day, the brushes were imaged using the procedure described above.

**Bacteria growth on HA films and glass substrates.** Post HA film preparation, 100 μL of the OD 0.02 culture was added to the sample and incubated at 37 °C. The sample area of the HA brush, HA film and glass was the same size for all cases. Imaging pre- and post-wash and the washing steps are identical to that for the biofilm.

**Bacteria quantification.** The total volume of living bacteria stuck to the surfaces in PAO1 biofilms after washing was quantified. Confocal microscopy was used to image a 3D volume. The total number of bacteria was determined for all samples by counting all voxels in the confocal z-stacks that had a signal that was above the noise threshold, indicating bacteria were present. Briefly, the 3D z-stacks were exported as .oib files in order to preserve the metadata and processed with Matlab through custom codes. The BioFormats package was used for reading the images and defining the stack structure for further processing. Noise reduction was performed using high-pass filtering with a Gaussian filter and the images were further binarized using a suitable thresholding algorithm. The total volumes of live bacteria on the brushes, films and glass substrates were calculated by counting the nonzero voxels in the binarized image stack. The average live bacterial volume in the region with a depth of 5 μm above the brush, film, and glass surface was used in calculations.

**Reporting summary.** Further information on research design is available in the Nature Research Reporting Summary linked to this article.

## Data availability

Raw data for all figures in the manuscript and the supplemental materials are provided online at https://github.com/wwb203/HyaluronanNatureCommunication. All data are available from the corresponding authors upon reasonable request.

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

## Acknowledgements

We are grateful to Prof. Andres Garcia for providing MEF cell line and Prof. Stephen Diggle for providing the fluorescent PAO1 bacteria. We also thank Prof. Blair Brettmann for helpful input and Shlomi Cohen and Yu Jing for help with cell culture. We gratefully financial support from the NSF DMR #0955811, #1709897 and NSF PoLS SRN # 1205878.

## Author contributions

W.W. was responsible for concept and research design, spherical brush experiments, fibroblast experiments, data interpretation, and paper preparation. J.L.F. was responsible for research design, planar brush experiments, micropatterning experiments, data interpretation, and paper preparation. H.S. was responsible for the biofilm experiments concept, research design, and data interpretation. D.T.K. was responsible for brush development and paper preparation. J.T. was responsible for the SEM experiments and data interpretation. F.R., E.R. and A.R.H. were responsible for the solid-state nanopore experiments and data interpretation. A.T.M. and M.A.F. were responsible for XPS experiments and data interpretation. M.K. was responsible for the AFM experiments and data interpretation. J.L.W. and P.H.W. were responsible for the development of HAS-rich *E. coli*, HAS fragments, data interpretation, and paper preparation. J.E.C. was responsible for concept and research design, data interpretation, supervision, and paper preparation. All authors read and approved the final manuscript.

## Competing interests

The authors declare no competing interests.
