## [Peer Review File · Nature Communications]

Reviewer #1 (Remarks to the Author):

This is an interesting paper with unexpected results, and the experiments were rigorously carried out. I found it an enjoyable paper to read and was struck by the exquisite control over HA brush growth the authors demonstrate and by the many methods they use to characterize the brushes and the facile manipulation of the brushes. Consequently, I think it is worthy of publication in Nature Comm.

We are thrilled that you enjoyed reading the manuscript and that you felt the experiments were thorough. Thank you very much for your time and feedback. We have addressed your comments and questions below with care.

I only have a few minor comments:

—The 20 nm particles penetrate the brush, with approximately 30% of the bulk concentration value reaching the surface. Can the authors comment on why the 20 nm particles are not seen in the brush itself? Presumably, each picture is a snapshot in time and there should be some NPs diffusing through the brush at any given instant.

The plot in Figure 2c shows that the intensity of the 20nm particles decreases as the measurement proceeds from the edge of the brush to the large sphere surface) until its lowest point of 30% and then it jumps back up to 80% intensity. We know from washing planar surfaces that the 20nm beads stick to the HA brush/glass interface - hence the peak at the large sphere surface. Despite this fact, the trend of decreasing penetration with increasing particle size is demonstrated. In the corresponding image, if you zoom in, you can see that there is faint red in the brush area which diminishes towards the sphere. The data were taken by azimuthally averaging around the beads to extract an average intensity at each point using such images. Hence, the intensity is there - it is just difficult to see.

—Based on the low and high ionic strength data, authors should use polymer physics theory and see how the swelling data fits the theory for a polymer brush in a good solvent. This may provide some new insights on the properties of the HA brush

Regarding this request: we agree this is an interesting proposition. To gain adequate insight and compare with theoretical predictions, one has to perform a systematic study of brush response to different ionic strengths – such work is significant enough that we feel it is beyond the scope of this manuscript. This work is underway and should eventually appear in a future publication.

—With every cycle of on-demand synthesis, there is a trend to a decrease in height of the brush. Can the authors comment on this? Could some of this be due to irreversible loss—untethering— of the brush for the surface.

We agree that there is a slow decay in the height of the brush after multiple synthesis events. Untethering should not be relevant here, since we purposely remove as much as possible of the HA with each round of hyaluronidase. However, we do have a reasonable hypothesis for why

this effect takes place. It is described in the following text, which has also been added to the manuscript:

The small decay in brush height after each regeneration step may be due to the hyaluronidase sticking to the surface. We found that we had to minimize hyaluronidase adsorption to the surface using BSA, or no brush would grow. It is possible that small amounts of hyaluronidase still bind to the surface, increasing with each exposure. These experiments demonstrate the potential for HA synthase brushes to be employed as regenerative interfaces.

Reviewer #2 (Remarks to the Author):

This work reports the growth of ultra-thick hyaluronan polymer brushes from surface bound HA synthase. The assembly of the HA synthase at flat and curved interfaces is first studied and some evidence for brush growth is presented. The ability of the resulting brushes to sequester nanoparticles is studied using confocal microscopy. The responsiveness of brushes to ionic strength is then characterised and the ability to regenerate surfaces after brush cleavage is demonstrated. Finally, the authors propose the application of such interfaces for the patterning of HA (although it is not clear in which context this would be applied) and to regulate the adhesion of eukaryotic and prokaryotic cells. Overall, this is novel interesting work, but characterisation is often incomplete and does not fully support some of the conclusions. This work should be publishable after additional characterisation and significant revisions.

We are grateful to the reviewer for his/her critical review of the manuscript and his/her appreciation of the effort as novel and interesting. The below feedback has led to a significantly improved version of the original manuscript, and we have taken care to address each comment thoroughly and in many cases included additional experiments.

Specific comments:

1. The characterisation of the surface chemistry and the molecular weight of HA brushes generated is incomplete. The surface chemistry of the resulting interfaces should be characterised by complementary techniques such as XPS and ToF-SIMS to confirm the nature of the HA grown. Only nanopore sizing was reported. This characterises the molecular weight only.

Our understanding of this comment is that you are asking for us to confirm that the resultant polymer in the brushes is in fact chemically, hyaluronan (HA). To summarize our proof for this, we have provided four pieces of complementary evidence. First in the original work that demonstrated the use of HA synthase to make HA, it was confirmed that the enzyme produced polymers with equal amounts of the two sugar monomers used by the enzyme to make the HA - this was done using radioactivity assays.¹ Second, in our work we use bacterial hyaluronidase

¹ Kumari, Kshama, and Paul H. Weigel. "Molecular cloning, expression, and characterization of the authentic hyaluronan synthase from group C *Streptococcus equisimilis*." *Journal of Biological Chemistry* 272, no. 51 (1997): 32539-32546.

(*Streptomyces hyalurolyticus*) to digest the brushes quickly and completely. This enzyme is established to be highly selective for HA only and its digestion of a sample is generally accepted and frequently used as evidence that a sample is HA.² Similarly, we used a selective HA binding protein bound to GFP to fluorescently label the brushes to extract density profiles. The specific binding to the brush indicates that the polymer is HA. Last, and new since the last version, we performed XPS of the HA brush interface and provided data that is consistent with HA being present at the interface (see Supplementary Info Section IIc).

In addition, considering the importance of the molecular weight determination to the establishment of the claims (ultra-thick brushes) for such a novel system, additional characterisation of the molecular weight of HA should be provided. Importantly, size exclusion chromatography should be used (at least for HA generated from beads as the surface area should be sufficient to generate enough HA). In addition, light scattering should be used to determine the hydrodynamic diameter of cleaved HA chains.

Extensive work has been published by one of the co-authors on the paper (Dr. Paul Weigel) using SEC-MALLS to characterize the MW distribution of the HA produced by the HA synthase in bacterial membrane fragments used in this work. In that manuscript,³ characterization of the HA versus synthesis time was performed, confirming that high molecular weight HA is generated. The resultant data from that manuscript is reproduced here (Fig 6 from Dr. Weigel's 2006 manuscript cited above).

That manuscript shows that the molecular weight, M_w , increases from 0 to 8 hrs with a maximum average at 8 hrs $\sim 5.9 \pm 1$ MDa. Our work yields similar results, with the M_w at 8 hrs equal to 6.3 MDa for one sample and 4.6 MDa for another sample, yielding an average of $M_w \sim 5.4$ MDa. Or when binning it all together to find the weight averaged molecular weight ($N=3699$), we arrive at $M_w \sim 5.1 \pm 2.5$ MDa (st. dev). Hence, the results are consistent within the error (width of the distribution).

Fig. 6. Kinetic analysis of HA product size distributions is shown. Membranes containing seHAS were incubated at 30 °C with nonradiolabeled substrates, and HA product size distributions were determined by MALLS (A) or agarose gel electrophoresis (B) as described in Materials and Methods. (A) M_w values were determined at the indicated times in independent experiments using four different membrane preparations. Values are the means \pm SEM of at least triplicates. (B) Samples in lanes 1–6 were taken at 0.25, 0.5, 1.0, 2.0, 4.0, and 8.0 h, respectively. Lane 7 is a mixture of Select-HA and Mega-HA standard ladders ($\sim 3 \mu\text{g}$ total) ranging from 0.5 to 4.5 MDa as indicated.

² Shimana E, Matsumura G. Degradation Process of Hyaluronic Acid by *Streptomyces Hyaluronidase*1. *The Journal of Biochemistry* 1980, **88**(4): 1015-1023. Ohya T, Kaneko Y. Novel hyaluronidase from streptomyces. *Biochimica et Biophysica Acta (BBA) - Enzymology* 1970, **198**(3): 607-609.

³ Baggenstoss, Bruce A., and Paul H. Weigel. "Size exclusion chromatography–multiangle laser light scattering analysis of hyaluronan size distributions made by membrane-bound hyaluronan synthase." *Analytical biochemistry* 352, no. 2 (2006): 243-251.

The SEC-MALLS data are additional evidence that the HAS membrane fragments do indeed produce extremely long polymers. Further comparison with the nanopore data indicates the nanopore results are consistent and not unexpected, when compared to the extensive and thorough SEC-MALLS studies of the membrane fragments.

To further support the nanopore results, we have included an agarose gel electrophoresis assay performed on the same samples measured by nanopore. The gel was taken at the same time on the same samples used in the nanopore measurements that generated Figure 1f. The sample is compared with a control HA sample of known molecular weights. Although not quantitative, it is clear that the collected HA is clearly of a high molecular weight (>2.5 Mda at 4 and 8h). This is consistent with both the SEC-MALLS and the nanopore data. The AGE results are included here for your inspection.

Figure: HA samples previously prepared by Curtis et.al collected at different synthesis time points along with the HA-Molecular Weight ladder (HA-MW) as the reference. The first gel lane loaded with the HA-MW ladder consisted of mixture of equal volume of four quasi-monodisperse HA samples of 100kDa, 1000kDa, 1.5 MDa and 2.5 MDa provided by Hyalose, LLC. (Oklahoma City, OK). Columns corresponding to HA collected at 1 h, 2 h, 4 h, and 8 h are shown. (B1 and B2 refer to duplicate samples (batch 1,2)).

To provide some further insight regarding our efforts: We opted to work with the nanopore technology to characterize the HA distribution because there are inherent limitations in measuring polydisperse high molecular weight polymers greater than a few MDa for methodologies like SEC, light scattering, gel electrophoresis, etc. Further, despite quite some effort, we were unable to gain access to a SEC-MALLS setup like that used by our now retired co-author, Dr. Paul Weigel. Additionally, having a detailed molecular weight distribution that literally involves counting the molecules will be spectacular input data for some of the future

work we hope to do examining predictions from polymer physics with the structure of the brush itself; the detailed distributions will be very empowering, compared to data from SEC-MALLS or other methods.

A table has been added to the Supplementary Info (Table S1) giving a summary of the nanopore data versus time and analysis of the results, including M_n , M_w , width of the distribution, length of the average sized polymer (based on M_w), estimated hydrodynamic radius and the number of chains measured. The estimated hydrodynamic radius comes from the knowledge of the M_w and the established scaling of high molecular weight HA as $R_H \sim MW^{0.7}$.⁴

We would also like to emphasize that the nanopore assay technique has been validated in a detailed study published in Nature Communications.⁵ That study is published by our co-authors Adam Hall and colleagues.

Hence, in conclusion, we hope the reviewer agrees that the nanopore data together with the extensive record of SEC-MALLS data for HA synthase already published is sufficient to confirm our claim that indeed the HA in the generated brushes is very high molecular weight and that the nanopore data is not unexpected – it is consistent and indeed enables us to examine the ultra-high molecular weight molecules in the outliers that would otherwise be missed because of issues with passage through SEC.

2. The grafting density of the resulting brushes is not properly characterised. There is an interesting discussion that indicates that it should be about two orders of magnitude lower than for other typical synthetic brushes generated via controlled radical polymerisation (typically in the range of 0.1 to 1 chain/nm²). Considering the height of the brush, there is no dispute that what is generated is a brush (thicknesses in the micron range, when chain densities indicate spacings of tens of nm). However, considering the importance of grafting density to the physical properties of brushes and the central stage of the HA brush structure in the novelty of this work, grafting densities should be rigorously characterised. Specifically, it can be determined by measuring the dry thickness of the polymer brush coating, in combination with the molecular weight of the corresponding brush. Using the classic equation in the field (see: Langmuir 2007, 23, 5769–5778 for example; assuming a density of 1.3 g/mL, which I did not check for HA, but is often used for other polymers), I calculated that a 2 MDa HA brush would have a grafting density equal to $4 \cdot 10^{-4}$ times the dry

⁴ Cowman, Mary K., and Shiro Matsuoka. "Experimental approaches to hyaluronan structure." *Carbohydrate research* 340.5 (2005): 791-809.

⁵Rivas, Felipe, Osama K. Zahid, Heidi L. Reesink, Bridgette T. Peal, Alan J. Nixon, Paul L. DeAngelis, Aleksander Skardal, Elaheh Rahbar, and Adam R. Hall. "Label-free analysis of physiological hyaluronan size distribution with a solid-state nanopore sensor." *Nature communications* 9, no. 1 (2018): 1037.

thickness (in nm). So for a dry thickness of 1 μm , it would give 0.4 chains/nm². However, I doubt the dry thickness would be so high. HA has a relatively high persistence length and tendency to swell. Hence when HA is adsorbed (grafting to) to surfaces, the dry thickness can often be in the range of 2-5 nm, yet the hydrated thickness can be as high as 100-200 nm. Therefore, we could anticipate that, in the brushes presented the dry thickness could be in the range of 100 nm (or less since the membrane fragments only cover 30% of the surface) and the grafting density could be as low as 0.04 chains/nm². This would be low but just in the range of polymer brushes. However, the authors have calculated that the density of HA synthase would be in the range of 2100 molecules/ μm^2 . If we assume 100% efficiency in initiating an HA chain, this would result in a grafting density of 0.002 chains/nm². This is 1 order of magnitude lower than the density reported above. Strikingly, brushes generated via controlled radical polymerisation are typically achieved with 10% compared to the density of initiator monolayers. However, it could be that the dry thicknesses measured are even lower (10 nm for example), which would bring their grafting density in the same range as that calculated for the HA synthase density by the authors. It would still imply near 100% initiation, which would be very remarkable. Therefore, I believe that elucidating the exact grafting density is particularly important for this report.

We appreciate the reviewer's encouragement to further determine the grafting density. Using the suggested dry thickness method that is widely accepted in the literature, we estimated the grafting density to be in the range of 0.0074-0.0021 molecules/nm². Details and discussion provided below.

Atomic force microscopy has been performed on dried brushes (4 hr growth time, N=3). An average dry thickness of 12.5 ± 0.7 nm was measured by performing a scratch test (see Supplementary Fig. S3).

A dry brush can be considered to be a brush in a very poor solvent (air) where it is fully collapsed. In this case, the height of a polyelectrolyte brush is related to the grafting density according to the following relationship:

$$H = N \tau^{-1} d^{-2} b^3$$

Here H = dry brush thickness, N = number of monomers, b = monomer length (b = 1 nm for HA), d = spacing between polymers ($d^2 = 1/\sigma$), and t is the second virial coefficient indicating the

solvent quality ($\tau = 1$ for a poor solvent⁶). The value of N is an estimate due to the brush's inherent polydispersity. We can use the average value of the M_w , which yields $N=16,900$; or the M_n which yields $N=6050$. Hence, using a range of $6050 < N < 16,900$ and $H=12.5$ nm, we arrive at a grafting density range of $0.00074-0.0021$ molecules/nm², or equivalently, 740 HA/um² - 2100 HA/um². The upper range (associated with $N=6060$) is consistent with our crude estimate based on HAS density in the fragments. However, if only 50% of those fragments face upwards and produce HA, the number would be halved. Additionally is unclear how to deal with the partial coverage of the surface (~33%). Interestingly, the grafting density associated with the weight-averaged molecular weight is 35% of the predicted HAS density.

While the grafting density cannot be accurately determined/defined due to the heterogeneity of the system, presumably these order-of-magnitude estimated values allow for general comparison to synthetic brushes, and the effort is sufficient to address the reviewers requests within reason.

These results have been included in the manuscript and supplemental materials.

3. A low grafting density would be more consistent with the particle infiltration experiments reported. Indeed, only 200 nm particles were excluded from the brush core and 20 nm particles significantly penetrated. Dextran chains of 10 kDa (5 nm) are proposed to diffuse through completely. This is at odds with the diffusion of even small molecules and peptides in dense polymer brushes (the biofunctionalisation of polymer brushes with even low molar mass compounds was reported to be limited for brushes with grafting densities in the range of 0.5 chains/nm², see: *Macromolecules* 2011, 44, 6868-6874 or *Polym. Chem.* 2016, 7, 979-990). This needs to be discussed more extensively, after analysis of exact grafting densities.

The relatively low grafting densities reported here combined with the very polydisperse nature of the brush is consistent with the penetration of particles into the HA brushes. Compared with 0.5 chains/nm², we are 2 orders of magnitude lower in density. The literature cited by the reviewer further confirms that decreased grafting density facilitates penetration into brushes. Given our low grafting density, the penetration profiles we report in the paper are not unreasonable. Additionally, the particle penetration assays were performed on the spherical polymer brushes rather than the planar, creating a less dense barrier to particles. Last, it is notable that polydispersity in brushes also increases the extent to which particles can penetrate into brushes⁷.

To address these concerns, we included the following discussion in the text:

⁶ Ross, Richard S., and Phil Pincus. "The polyelectrolyte brush: poor solvent." *Macromolecules* 25, no. 8 (1992): 2177-2183.

⁷ Qi, Shuanhu. "Particle Penetration into Polydisperse Polymer Brushes: A Theoretical Analysis." *Macromolecular Theory and Simulations* 26, no. 5 (2017).

While traditional dense synthetic brushes can exclude molecules (Polym. Chem. 2016, 7,970-990), the partial penetration into these spherical HA brushes is not unexpected given the relatively low grafting densities, the spherical geometry, and the broad polydispersity – all of which are expected to enhance particle penetration into brushes.

4. “We studied the impact of brush loss from pipetting effects on 4hr growth brushes...” (p4). The wording of this section is a little odd. What is reported is simply molecular desorption. Or do the authors mean that pipetting introduces shear that disrupts the brush? If it is desorption, let’s call it this way.

We have clarified the text to indicate that we are examining the shear-induced desorption of HA from the enzymes, not that we are breaking up HA by just pipetting nor just measuring the natural desorption over time (with no shear). The language has been improved in the manuscript to clarify this point:

“We studied the impact of brush loss from pipetting effects on 4 h growth brushes, and found that the height change is consistent with loss from solvent exchange (shear induced desorption), rather than the natural desorption of HA from HA synthase over short time periods (Supplementary Fig. S7).”

And further in the supplemental materials:

“...The effect was to mimic conditions of exchanging solvents. Over the course of one hour and 18 gentle pipette pump actions, the average height of the planar brushes was relatively stable (Figure S10), showing a slight decrease in average values. This induced height decrease is more significant than for a brush measured every 20 minutes with no solvent swaps (see Figure 4d in manuscript). Hence shear induced desorption driven by pipetting increases brush decay relative to the decay from natural desorption.”

5. The crosslinking (reinforcement) of HA brushes should be better characterised. Since no amine crosslinker is introduced, it will rely on ester formation between HA and HA-NHS activated chains. This is usually relatively inefficient (NHS esters in the conditions used have only a half-life of 30 min and amine are orders of magnitude more reactive than alcohols with these activated groups). However, it could involve crosslinking to the PEI. Some surface chemistry characterisation is needed here. In this context, Figure S9 is not fully clear (the caption is not clear). The treatment with hyaluronidase (Figure S10) does not confirm that there is no “crosslinking between the HA strands” as hyaluronidase would also digest such crosslink brush.

We have removed the statement about the hyaluronidase and brush crosslinking.

The manuscript now clearly states the expected mechanism for the crosslinking to be between the PEI and the HA. We have improved the language for clarification:

“Thus, we developed a strategy to generate stabilized brushes by crosslinking the HA to the underlying PEI surface chemistry with EDC-NHS chemistry. The EDC activates the carboxyl groups on HA for conjugation to secondary amine groups available in the PEI layer of the underlying grafting surface.”

We were unable to make measurements to confirm this crosslinking - especially given the limitations of XPS when measuring such a complex interface (see discussion in Supplementary Info IIc).

After our recent experience with characterizing the interface with XPS to prove the polymer is HA to respond to an earlier request, and looking at how complex the multi-layered chemistry is, we found it would be difficult to prove the crosslinking using techniques accessible to us. The redundancy in the bonds that exist in the several chemical layers (PEI/glutaraldehyde as well as the bacteria fragments and HA) make the small changes in spectra expected with binding at the interface impossible to discern from before and after the reinforcement procedure is performed.

At the same time, the mechanism of the crosslinking expected is straightforward and we have provided abundant data that demonstrates that the crosslinking/brush reinforcement works for up to one year. Given that the purpose of the manuscript is to introduce and characterize the HAS enzyme fabricated ultra-thick brushes, and the fact that we have demonstrated the stability of the crosslinked brushes over months, we ask the reviewer to allow us to confirm the chemistry, if needed, in future work since it is not a crucial aspect to the main messages of the work.

6. The section on cell-brush interactions is incomplete. There is little evidence for HA digestion by MEFs. The absence of dextran stain (Fig 6b) is simply reflecting its exclusion by the cell body.

We have included previously missing evidence that the cells digest the HA brush as they adhere. This data was not included by oversight, and it was these missing results that influenced the original discussion presented in the text.

There are two pieces of direct evidence that HA is removed by the cells. The first is fluorescent images of the HA brush interface (at the glass) before and after exposure to cells for 12 hours (now Fig 6a). The images reveal distinct 'tracks' in the HA at the surface, where black regions are clearly visible. This indicates that HA has been fully removed from the surface in certain areas and despite the expected 'healing' property of the brush, since the extent of removal is significant enough to leave visible scars. This is indicative that HA has been substantially removed by the cells - since the examined brush is reinforced and HA should not detach in the 12 hour period of the experiments. This image is now in the main text in Figure 6A and included here for your ease (scale bar is 50 microns).

Furthermore, particle exclusion assays on the reinforced, cell exposed brushes (12 hrs) often revealed regions where the bead height (brush height) is very inhomogeneous. When focusing at the surface of the sample, no beads should be visible if the brush remains intact. Yet frequently, the beads were clearly visible, indicating that the brush had been compromised or removed. A figure showing this is now included in Fig 6B and several side views from the confocal stack in the Supplementary (Figs S15), as well as below for your ease (scale bar 50 microns in x-y, scale bar 20 microns on x-z sideviews). We discern that the destruction of the brush must be cell driven, since the reinforced brushes are stable over months.

Regarding dextran penetration under the cell - we have found that cells adhered to brushless surfaces (glass and/or bacterial membrane fragments with no grown brush) do not allow dextran to penetrate beneath the cell surface. A typical image has been included in Suppl. Fig S13b. Yet, for cells adherent to brushes, dextran does always penetrate under portions of the cell, sometimes with very large gaps (Fig 6b, 6c and Suppl. Fig S13c). This is still discussed in the manuscript but is no longer used as the main evidence that HA is digested by the cells.

The manuscript text has been substantially revised to address these updates and clarifications - changes have been made throughout the entire passage concerning brush-MEF interactions.

Cells may very well adhere to the brush itself. Indeed, fibroblast typically express HA receptors and these can mediate matrix assembly, therefore enabling cells to assemble focal adhesions. Note also that the absence of FAs underneath the cell body (where dextran is still localised) is not surprising as fibroblasts (in particular MEFs) typically generate FAs at protrusions and the edge of lamellipodia.

It is an excellent point that the cells could adhere via CD44 to the brush (or other HA receptors). We have included a comment mentioning this likely interaction. Further, we removed the language that suggests that the remaining HA under the cell is surprising.

7. There is no characterisation of cell viability, which would support some of the claims made.

Following the reviewer's excellent suggestion, the cell viability after 12 hours growth on the brush interface was surveyed using a live / dead cell assay. We found that out of ~3000 measured cells, only 6 were dead. The results combine measurements from three separate surfaces. These results have been added to the manuscript with typical images included in the Supplementary Materials.

8. In the antibacterial work, it is stated: "Few bacteria reached the underlying substrates in this time...". This makes assumption as to the mechanism of bacterial adhesion. Bacteria may be able to reach spaces where membrane fragments were absent (and that would be positively charged). Bacteria may also manage to adhere to the brush. High resolution confocal microscopy would help confirming which scenario is prevailing.

The data included in the original manuscript is from 3D high resolution confocal microscopy. We examined 3D stacks of confocal images for the presence of bacteria within the biofilm or at the surface before and after the washing steps. We do not make any assumptions regarding mechanisms of bacterial adhesion, or at least we do not intend to in the way the manuscript is written. But we did verify that it is rare for the bacteria to reach the surface of a brush, and further, that despite being in solution above the HA brush for extended periods, the majority of bacteria are easily removed with no sign of an adherent biofilm.

9. MEF adhesion to reinforced brushes has not been characterised. If such coatings are to be useful for tissue engineering (to promote cell adhesion whilst resisting bacterial adhesion), reinforced brushes need to display good cell adhesion and culture performance whilst preventing bacterial adhesion. This has not been demonstrated.

The data shown in Figure 6 (originally) and now in our follow up experiments of live/dead cell assays were all collected using reinforced brushes. We have updated language in the manuscript to clarify this.

10. Why quantify bacterial volumes rather than simply counting the number of bacteria per surface area (which is more straightforward)? Fig 6g-i do not seem to match, possibly due to the way the bacterial densities have been characterised. For example, bacterial densities post wash on HA film and HA brushes seem comparable on the image, but appear very different in Figure 6i. In addition, showing the relative percentages does not make much sense. If the range of bacterial adhesion is broad and the authors want to highlight the high and low range, a logarithmic scale could be used.

We have updated the values to report number of bacteria rather than volume in the Table S3. Further, the figure now reports the total bacterial count rather than relative percentage on a logarithmic scale (great suggestion, thanks!).

The difference in the images in Figure 6 has been adjusted for more straightforward interpretation. In the old data showing the brush image, we included blue color for dextran, as we did for all brush images. That may have been confusing and made the images of the film and the brush look similar when they were not; the dextran coloring has been removed. The final images now can clearly be distinguished.

11. In p6, the authors state “The enzyme is able to polymerize much higher molecular weight molecules than other techniques such as free radical polymerization”. This is not strictly speaking true. Although I do not dispute the validity of this assertion for most free radical and controlled radical polymerisations, recent work on SET-LRP (see work by Percec) has resulted in the growth of polymer chains in the range of several 100 kDa to MDa.

Thank you for bringing this to our attention. We have adjusted the statement to specify we mean in the context of polymer brushes, and added a reference to SET-LRP^{8,9}

The enzyme is able to polymerize much higher molecular weight molecules than most other techniques such as free radical polymerization with few exceptions (Vorobii, Mariia, et al. Polymer Chemistry 6.23 (2015): 4210-4220.; Zoppe, Justin O., et al." Chemical reviews 117.3 (2017): 1105-1318.)

12. P6, “The system enables the high-resolution visualization of brushes for the first time”. Strictly speaking, the resolution has not improved. It is the brush that is larger.

We agree, this should be reworded to be more precise.. We have changed the sentence to say:

“...enables the direct visualization of the detailed spatial structure of brushes for the first time.”

13. “for example stopping after small growth, and then continuing growth after labelling...”. End chain functionalisation of polymer brushes generated via ATRP has been reported previously.

We removed the words at the end of the sentence: *‘that were previously inaccessible’*.

14. There is at least one reference missing (p3): “on their distribution (REF).”

This has been updated.

15. There are too many typos, grammatical errors, words missing etc. The manuscript needs a complete proof read and substantial improvement in the style. In addition, units are often directly reported after the value (25nm), when there should be a space. Finally, the correct abbreviation for minutes and hours is min and h, not mins and hrs or hr. This should be corrected throughout.

These issues have been corrected.

⁸ Vorobii, Mariia, Andres de los Santos Pereira, Ognen Pop-Georgievski, Nina Yu Kostina, Cesar Rodriguez-Emmenegger, and Virgil Percec. "Synthesis of non-fouling poly [N-(2-hydroxypropyl) methacrylamide] brushes by photoinduced SET-LRP." *Polymer Chemistry* 6, no. 23 (2015): 4210-4220.

⁹ Zoppe, Justin O., Nariye Cavusoglu Ataman, Piotr Moczyn, Jian Wang, John Moraes, and Harm-Anton Klok. "Surface-initiated controlled radical polymerization: state-of-the-art, opportunities, and challenges in surface and interface engineering with polymer brushes." *Chemical reviews* 117, no. 3 (2017): 1105-1318.

16. Some of the abbreviations are introduced without explaining what they mean. For example, UDP-GlcUA is introduced without stating the full name of the chemical in the text (it is in the methods and the SI though).

We have addressed this issue, and identified and fixed any other places where abbreviations are not properly introduced.

17. “Arena” is used throughout instead of “area”. Although it can be used in this way occasionally, it is not always appropriate.

We have identified the two places in the manuscript where ‘arena’ is used and feel that in both cases its implementation is appropriate. Further, using ‘arena’ prevents confusion with the scientific term ‘area’ that is used throughout the paper.

Reviewer #3 (Remarks to the Author):

This manuscript reports on the synthesis and characterization of ultra thick hyaluronan brushes for potential biomedical applications. The strategy is quite different from the classical grafting to and grafting from method. Namely, the authors first immobilized hyaluronan synthase on flat or curved surfaces, and grow the brush from the surface (similar to grafting from method). The thickness of the brush reached an impressive 20 micron value. The authors also demonstrated that patterned brushes and “programable” growth can be accomplished.

This work is quite interesting, approaching a classical polymer problem (growing polymer brushes) using a matured biochemical method. It is novel and could be potentially published in nature commun, after addressing the following questions:

Thank you for your supportive feedback and thoughtful comments. We have addressed them below.

1. The manuscript was presented more from a biochemistry stand point, due to the method used. However, since the targeted problem is on polymer brushes, it is necessarily to incorporate standard “polymer language. In terms of polymer brush characterization, the grafting density argument was on hyaluronan synthase (or “initiator”). The final chain grafting density should be measured and discussed.

In response to this request and that of reviewer 2, we measured the grafting density using a more standard approach from polymer brush chemistry. First we dried the brushes, then we used AFM to measure the dry height (~12.5 nm). From that, we were able to back out the grafting density. As reported in detail in the text, and also described to reviewer 2, we found that using this approach, the grafting density is between 740-2100 chains/ μm^2 or rather in more standard polymer language, 0.00074-0.0021 chains/ nm^2 .

2. Similar question, on page 4, the authors pointed out that chains can be removed by washing/solvent exchange. Again, based on standard polymer brush study, this indicates

loose chains. They are typically removed by vigorous washing/sonication before the grafting density is measured.

This is a valid point. However, we are transparent about our measurement methods and very open and in fact quantify chain loss. Given the uniqueness of this system, with the unreinforced brushes being attached very differently than synthetic polymer brushes, and the fact that if we washed long and hard enough, all of the chains could be removed, we believe it makes sense to leave the work as it is. Moving forward, in future efforts, we will be more aware of this standard, and typically default to using brushes that have been reinforced by covalent bonding to the surface.

3. The main indicator for the brush is the high in wet state. While this perhaps is most relevant to potential biomedical applications, it's not clean/sufficient from a polymer brush standpoint. Dry state height and the related grafting density should be measured and compared with other existing polymer systems.

See response to request 1 above - we have done this and inserted the results into the manuscript. The height is 12.5 +/- 0.7nm (n=3 measurements) and the estimated grafting density is 0.00074-0.0021 chains/nm².

4. What's the molecular weight of the brushes?

Currently we have the molecular weight of the samples produced by the HAS fragments in solution (Fig 1f), but not the molecular weight of the brushes themselves. Since receiving this review, we have attempted several times to acquire data for the brushes but have not yet succeeded. The failure is due to combination of gathering enough sample (there is much less coming from the small area interfaces we prepare) and having access to nanopore devices to perform the measurements. Although it is not ideal, we hope the reviewer can agree that the molecular weight distribution for the HAS fragments is sufficient for this primarily qualitative introduction of the HA brush platform. Future work will definitely address (and require) detailed information of the HAS brush MW distribution.

The nanopore assay data shown in Fig 1f gives the molecular weight distribution versus time as produced by the HA synthase in the membrane fragments. A table summarizing the number averaged and weight averaged molecular weights and the corresponding PDI has been added to the Supplementary Info (Table S1).

We can also estimate something about the average effective contour length by studying the brush height in very low salt conditions, when the brush stretches to 87% of its contour length (as shown for monodisperse HA brushes by Richter et al.¹⁰ In pure water, the brush height after 4 hr growth is ~10 microns, which means that the effective average contour length is $10/0.87 = 11.5$ microns. $1 \text{ nm} = 400 \text{ Da}$, so this corresponds to an effective average molecular weight estimate of 4.6 MDa. This is not so different from the values

¹⁰ Attili, Seetharamaiah, Oleg V. Borisov, and Ralf P. Richter. "Films of end-grafted hyaluronan are a prototype of a brush of a strongly charged, semiflexible polyelectrolyte with intrinsic excluded volume." *Biomacromolecules* 13, no. 5 (2012): 1466-1477.

acquired from the fragment data, which have now been summarized in the Supplemental Info in Table 1.

5. Page 3, regarding brush thickness on spheres, it is understandable that under similar condition, brush height is smaller on a sphere surface. However, why does the height plateaus at such an early stage of growth (8 hrs)?

Good question; this clearly requires more investigation. The default explanation is that the polymers are still growing on the spherical particle after 8 hrs (or perhaps the distribution is changing but the peak length has been reached), since the thickness doesn't appear to quite plateau at that time. However, at 8 hours, at those heights in the spherical brush, there is so much available space that the polymers do not stretch out in brush-like formation, and so the increase in height is negligible compared to the planar brush.

6. Page 3, paragraph 3, a reference is missing.

Inserted, thank you.

REVIEWERS' COMMENTS:

Reviewer #1 (Remarks to the Author):

The authors have responded adequately to my queries, and have also carried out experiments to address some of the issues raised by the other reviewers. Consequently, I have no further concerns.

Reviewer #2 (Remarks to the Author):

The authors have addressed all of the comments previously raised and this manuscript is now nearly ready for publication. Really interesting work! Only two aspects still require minor corrections:

1. In Figure S5: the caption indicates a panel f that should present the C1s XPS trace of brushes grown for 8h. I cannot see this data.
2. Figures 6a/b do indicate digestion of the brush by cells, but we need to see comparable images prior to cell adhesion (not necessarily for the same areas, although this would be more striking). The authors mention this in their reply, but I cannot find this material.

Reviewer #3 (Remarks to the Author):

The authors have addressed original questions raised by the reviewer, and the revised manuscript can be accepted for publication.

REVIEWERS' COMMENTS:

Reviewer #1 (Remarks to the Author):

The authors have responded adequately to my queries, and have also carried out experiments to address some of the issues raised by the other reviewers.

Consequently, I have no further concerns.

Great – thank you!

Reviewer #2 (Remarks to the Author):

The authors have addressed all of the comments previously raised and this manuscript is now nearly ready for publication. Really interesting work! Only two aspects still require minor corrections:

1. In Figure S5: the caption indicates a panel f that should present the C1s XPS trace of brushes grown for 8h. I cannot see this data.

Done. Edited the caption to match the figure.

2. Figures 6a/b do indicate digestion of the brush by cells, but we need to see comparable images prior to cell adhesion (not necessarily for the same areas, although this would be more striking). The authors mention this in their reply, but I cannot find this material.

We have added a x–y slice of the fluorescently labeled brush to the supplemental materials (Supplementary Figure 13). Unfortunately this is not the same area as that which was imaged with the cells. However, we have imaged multiple areas of the fluorescently labeled brush and it is uniform with no such structure as seen when cells are present.

Reviewer #3 (Remarks to the Author):

The authors have addressed original questions raised by the reviewer, and the revised manuscript can be accepted for publication.

Thank you!